

**Quantifying late-Holocene climate in the Ecuadorian Andes using a chironomid-based temperature inference**
**model**
Frazer Matthews-Bird[1&2], matthewsbirdf@fit.edu
Stephen J. Brooks[3], S.Brooks@nhm.ac.uk
Philip B. Holden[1], philip.holden@open.ac.uk
Encarni Montoya[1], encarni.montoya@open.ac.uk
William D. Gosling[1, 4], W.D.Gosling@uva.nl
[1]Department of Earth, Environment & Ecosystems, The Open University, Walton Hall, Milton Keynes, MK7 6AA,
UK.
[2] Biological Sciences, Florida Institute of Technology, 150 West University Boulevard, Melbourne, FL 32901, USA
[3]Department of Life Sciences, Natural History Museum, Cromwell Road, London SW7 5BD, UK.
[4]Palaeoecology & Landscape Ecology, Institute for Biodiversity & Ecosystem Dynamics, University of
Amsterdam, P.O. Box 94248, 1090 GE Amsterdam, The Netherlands
**Corresponding author:** Frazer Matthews-Bird
**Key words:** Bayesian, weighted-averaging, transfer function, chironomids, Holocene climate change, Ecuador





**Abstract**

Presented here is the first chironomid calibration dataset for tropical South America. Surface sediments

were collected from 59 lakes across Bolivia (15 lakes), Peru (32 lakes) and Ecuador (12 lakes) between 2004 and
2013 over an altitudinal gradient from 150 m above sea level (a.s.l) to 4655 m a.s.l, between 0-17°S and 64-
78°W. The study sites cover a mean annual temperature (MAT) gradient of 25°C. In total, 55 chironomid taxa
were identified in the 59 calibration data-set lakes. When used as a single explanatory variable, MAT explains
12.9% of the variance ($\lambda_1/\lambda_2$= 1.431). Two inference models were developed using weighted averaging and
Bayesian methods. The best performing model using conventional statistical methods was a WA (inverse) model
($R^2_{jack}$= 0.890, RMSEP$_{jack}$ = 2.404, Mean bias$_{jack}$= -0.017, Max bias $_{jack}$=4.665). The Bayesian method produced a
model with $R^2_{jack}$= 0.909, RMSEP$_{jack}$ =2.373, Mean bias$_{jack}$= 0.598, Max bias $_{jack}$= 3.158. Both models were used to
infer past temperatures from a *c.* 3000 yr record from the tropical Andes of Ecuador, Laguna Pindo. Inferred
temperatures fluctuated around modern day conditions but showed significant departures at certain intervals
(*c.* 1600 cal yr BP; *c.* 3000-2500 cal yr BP). Both methods (WA/Bayesian) showed similar patterns of
temperature variability; however, the magnitude of fluctuations differed. In general the WA method was more
variable often inferring unrealistically cold temperatures (*c.* -7±2.5°C relative to the modern). The Bayesian
method provided temperature anomaly estimates for cool periods that lay within the expected range of the
Holocene (*c.* -3±3.4°C). The chironomid-based MAT reconstruction from the Laguna Pindo fossil record suggests
that periods of low solar output not only affect the tropics through changes in precipitation, but also directly
affect tropical temperatures. Inferred temperatures were 2-3°C colder relative to the modern during the widely
recognised 3500-2500 cal yr BP cooling event. Long-term cooling during the late-Holocene culminating in the
Little Ice Age (LIA) is not apparent in the Laguna Pindo record. A cooling by 1-2°C relative to the modern during
the LIA is recorded in a single fossil sample.







## 1. Introduction

Holocene climate variability (11.7 kcal yrs BP – present) offers the most recent opportunity to parameterise climate and ecosystem responses to natural forcing under current boundary conditions in the absence of intense anthropogenic activity (Mayewski *et al.*, 2004; Oldfield and Steffen, 2014). Furthermore, quantitative estimates of past climate over long time scales (>1000 yrs) are vital to improving the reliability of modelling and prediction of present and future climate variability (Mayewski *et al.*, 2004). The spatial distribution of palaeoclimate records, however, is currently uneven around the world. Quantitative reconstructions of past climate are common from mid- to high- latitudes of both hemispheres but data is much scarcer from low-latitude (tropical) regions (Jansen *et al.*, 2007). Tropical climate is the dominant driver of atmospheric circulation (Ivanochko *et al., 2005)* and the source of intermittent phenomena, such as the El Niño Southern Oscillation (ENSO), which has a global influence on climate (Collins *et al.*, 2010). Quantitative estimates of past climate from the low latitude tropics, therefore, are crucial for investigating not only regional climate processes, but also teleconnections on long timescales (>1000 years) (Garreaud *et al.*, 2009; Jomelli *et al.*, 2009; Vuille *et al.*, 2000). Here we develop the first chironomid-based temperature inference model for tropical South America. The model is applied to a Holocene lake sediment sequence to generate the first quantitative chironomid-inferred temperatures from the tropical East Andean flank.

Chironomidae (non-biting midges) is a family of two-winged aquatic insects of the order Diptera. The family is globally distributed and one of the most diverse within aquatic ecosystems (Armitage *et al.*, 1995). Many species are stenotopic, and their short life-cycles and ability to colonise favourable regions quickly means the insects are extremely sensitive to environmental change (Pinder, 1986). The head capsules of chironomid larvae are well preserved in lake sediments and have been used extensively as palaeoecological proxies (Brooks, 2006; Walker and Cwynar, 2006). Chironomid-based temperature inference models, derived from modern calibration data sets, have been applied across North America (reviewed in Walker and Cwynar, 2006), Eurasia (reviewed in Brooks, 2006), and more recently the method has been applied in the Southern Hemisphere in Patagonia (Massaferro and Larocque, 2013; Massaferro *et al.*, 2014), Central America (Wu *et al.*, 2014), East Africa (Eggermont *et al.*, 2010), and Australasia (Dimitriadis and Cranston 2001; Woodward and Shurlmeister 2006).



Transfer functions make a number of underlying assumptions; particularly the environmental variable to be

reconstructed is an ecologically important determinant in the system, and environmental variables other than

one being reconstructed have a negligible effect on species assemblages (Juggins, 2013). Rarely are ecological

systems as simple as transfer functions would imply and violations of these assumptions will undermine the

validity of the environmental reconstruction (Juggins, 2013). Nevertheless, despite known inherent problems

associated with transfer functions (Huntley, 2012; Juggins, 2013; Velle *et al.*, 2010), quantitative reconstructions

from chironomid assemblages often produce consistent results that compare well with other proxy estimates of

past temperature (Brooks, 2000; Brooks *et al.*, 2012; Heiri *et al.*, 2007). The best performing inference models

can reconstruct temperatures with errors of *c.* 1°C (Brooks and Birks, 2001; Eggermont et al., 2010; Heiri et al.,

2003; Olander et al., 1999a; Rees et al., 2008; Self et al., 2011) providing high resolution insights into past

changes in climate (Brooks and Langdon, 2014), and validation of climate models (Heiri *et al.*, 2014).

### 1.1 Holocene climate variability

Holocene climate variability is subdued (±2-3°C) (Mayewski *et al.*, 2004; O'Brien *et al.*, 1995) compared with

the preceding Late Glacial period (*c.* 25,000-11,700 years before present [yrs BP], ±7-10°C) (Alley, 2000;

Anderson, 1997), nevertheless rapid climate change events are recognised in Holocene palaeoclimate records

(Mayewski *et a*l., 2004). Changes in insolation caused by solar forcing is generally regarded as the dominant

driver of global climate change during the Holocene (Mayewski *et al.*, 2004; Wanner *et al.*, 2008). The Roman

warm period (250 BC-400 AD [2200-1550 yrs BP]), and cooling during the Little Ice Age (LIA) (1350-1850 AD

[600-100 yrs BP]) are well established features, notably across the Northern Hemisphere (Johnsen *et al.*, 2001;

O'Brien *et al.*, 1995). Growing evidence from the tropics suggests Holocene climate fluctuations such as the LIA

are probably global events (Thompson *et al.*, 2002; Wanner *et al.*, 2008); however, additional quantitative
palaeoclimate records are needed to understand the expression of such events in the tropics, and to clarify
global climate teleconnections. Although the low latitudes receive 47% of planetary insolation, the climate
response in the tropics to solar variability is poorly understood (Crowley, 2000; Polissar *et al.*, 2006).




### 1.2 Holocene climate variability in tropical South America

The most notable feature of current South American climate is the annual migration of the Intertropical Convergence Zone (ITCZ), which affects rainfall patterns across the tropical Andes (Abbott *et al.*, 2003; Bird *et al.*, 2011; Haug *et al.*, 2001). On Holocene timescales, however, there remain large uncertainties regarding the patterns and processes of climate change in the Andes with evidence for both rapid (*c.* 100-1000 yr) precipitation (Haug *et al.*, 2001) and temperature variability (Thompson *et al.*, 2006; Wanner *et al.*, 2008). A further point to note is the spatial heterogeneity of Holocene climate variability in the tropical Andes (Baker and Fritz, 2015a), particularly regarding precipitation. Ice core records from the Peruvian and Bolivian Andes since *c.* 5400 cal yrs BP suggest the overall trend is towards a drier climate with high amplitude fluctuations and periods of significant aridity. Precipitation reached a minimum during the period between 3800-2800 cal yrs BP and the LIA (Haug *et al.*, 2001; Thompson *et al.*, 1986; Thompson *et al.*, 1995). Speleothem records from the Central Andes of Peru contradict this, however, and indicate instead that from the 15[th] to 18[th] century precipitation was on average about 10% higher than the present day (Reuter *et al.*, 2009).

The mid- to late-Holocene (*c.* 6000 cal yrs BP to present) is a period of cooling climate in South America. Pollen evidence suggests montane vegetation replaced Andean forest taxa as the treeline lowered with modern vegetation patterns becoming established by *c.* 3000 cal yrs BP (Markgraf, 1989). Long-term cooling in the late Holocene culminated in a minimum during the 17[th] and 18[th] centuries, coinciding with evidence for precipitation minimum during the LIA in northern South America (Haug *et al.*, 2001; Thompson *et al.*, 1986; Thompson *et al.*, 1995). Further south, Patagonian proxy records infer periods that were wet and cold enough to allow glacial advance (Meyer and Wagner, 2008).

Stable isotopic records derived from ice cores and speleothems provide some of the highest resolution archives of South American palaeoclimate (Kanner *et al.*, 2013; Mosblech *et al.*, 2012; Thompson *et al.*, 1986; Thompson *et al.*, 1995). These records must be viewed cautiously, however, as the proxy also reflects a range of environmental factors; changes in water vapour, snow surface histories, source and sink ratios, and the effect of temperature on carbonate-water isotopic fractionation (Baker and Fritz, 2015; Leng and Marshall, 2004; Thompson *et al.*, 1986). Without solid constraints on temperature variability through time it is difficult to be certain of the isotopic signal. In the South American tropics, where the relationship between changes in



temperature and precipitation are complex (Baker *et al.*, 2001; Garreaud *et al.*, 2009), more independent
quantitative estimates of past temperature are needed in order to resolve climate patterns over the tropical
Andes during the Holocene.

**1.3 Aims**
In this study, we have developed the first chironomid-based temperature calibration data set from the
tropical Andes (0 to 17°S). Surface sediment samples from 59 lakes along the eastern flank of the Andes to
Amazonia are analysed. Two approaches are used to develop the inference model, the widely used weighted
averaging method (Brooks and Birks, 2000) and a second using a Bayesian approach (Holden *et al.*, 2008) which
has rarely been used before. The models are applied to fossil chironomid assemblages in a late-Holocene lake
sediment record from Laguna Pindo, central Ecuador, to reconstruct mean annual temperature (MAT) changes
over the past *c.* 3000 years.

**2   Study Sites**
**2.1 Modern calibration dataset**
Surface sediments were collected from 59 lakes across Bolivia (15 lakes), Peru (32 lakes) and Ecuador (12
lakes) between 2004 and 2013 over an altitudinal gradient from 150 m above sea level (a.s.l) to 4655 m a.s.l,
between 0-17°S and 64-78°W (Fig 1). The study sites cover an MAT gradient of 25°C; the coldest lake in the data
set is 0.8°C MAT and the warmest is 25°C MAT (Table 1). The deepest lake is 25 m and the shallowest is 0.1 m,
mean water depth of all the study sites is 5 m. Cold, high elevation lakes are more common within the
calibration data set and there are no lakes between 16°C and 20°C. Sediment samples used in this study were
taken from the uppermost centimetre (0-1cm) which represents the most recent deposits (approx. 5-20 years)
(Frey, 1988) and therefore most comparable with the available climate data for calibration.





**2.2 Fossil chironomid record**
Laguna Pindo is a small shallow lake on the eastern flank of the Ecuadorian Andes (1°27.132'S;
78°04.847'W) (Fig1). The site is located at an elevation of 1248 m a.s.l. MAT is *c.* 20°C with little seasonal
variation and mean annual precipitation (MAP) can reach *c.* 4000 mm per year (Hijmans *et al*., 2005). Currently
the lake is not directly fed by a stream in-flow and has no visible stream out-flow; the lake receives water from
surface run-off and direct precipitation. There are no obvious geomorphological causes for the escarpment of
the lake and we hypothesise it is tectonic in origin.
At the time of field work (January 2013) maximum water depth was *c.* 1 m, the lake is heavily overgrown
with aquatic macrophytes making a detailed bathometric survey difficult. A sedimentary sequence 929 cm long
was extracted using a cam-modified piston Livingston corer (Colinvaux *et al*., 1999) from the centre of the lake
to minimise the chance of encountering a sedimentary gap caused by any periods of lake area reductions.
Sediments were recovered in aluminium tubes and sealed on site before being transported to the UK and
stored at *c.* 4°C. A total of 14 samples were analysed for $^{14}$C radiocarbon using AMS dating at the SUERC
radiocarbon facility, East Kilbride. An age-depth model was created using version 2.2 of the statistical package
clam.R (Blaauw, 2010) and the Southern Hemisphere calibration curve SHCal13.14C (Hogg *et al.*, 2013).

**3. Methods**
**3.1 Chironomid analysis**
Chironomid preparation and identification from both lake surface and core sediments followed standard
methods as described by Brooks *et al* (2007). The wet sediment was deflocculated in 10% KOH for 2 minutes at
75°C. The sediment was then washed through 212μm and 90μm sieves with water. Chironomids were picked
from the residues in a Bogorov counting tray using a stereomicroscope at 25x magnification. Head capsules
were mounted in Euparal, ventral side up and identified to the highest possible taxonomic resolution under a
compound light microscope at 200-400x magnifications with reference to Wiederholm (1983), Epler (2001)
Rieradevall & Brooks (2001), Brooks *et al* (2007), Cranston (2010) and local taxonomic works including Prat *et al*
(2011), and Trivinho-Strixino (2011). Some taxa could not be formally identified and so were given informal
names. Images and descriptions of informally named taxa are provided in Matthews-Bird *et al* (2015).




### 3.2 Environmental variables


Environmental variables (depth, pH, conductivity, and water temperature) were measured at each lake in
the field. Organic content of the sediment was established through Loss-on-ignition following standard methods
as described by Heiri et al (2001). Climate data (MAT, MAP) were obtained from high resolution, interpolated
climate surfaces (Hijmans *et al*., 2005). Elevation (m a.s.l.), latitude (decimal degrees), and longitude (decimal
degrees) were also included as variables in the calibration data set. A summary of all the environmental
variables measured can be found in Table 1.


### 3.3 Exploratory statistics


Detrended Correspondence Analysis (DCA) was initially used as an indirect ordination method to assess the
gradient lengths in compositional units of taxon turnover (Hill and Gauch, 1980). The gradient length of DCA axis
1 was 5.2 standard deviation units (SD), which suggests a unimodal response, and that linear ordination
methods were not appropriate (ter Braak, 1987). Canonical Correspondence Analysis (CCA) was used to explore
the influence of the measured environmental variables on the distribution and abundance of taxa. Highly
correlated variables were partialled-out by analysis of the variance of their regression coefficients indicated by
their Variance Inflator Factors (VIFs). Variables with high VIFs were systematically removed from the
environmental variable data set until the remaining variables had VIFs below 20. Detrended canonical
correspondence analysis (DCCA) was used to test how much of the variance in the assemblage data was
explained by each individual explanatory variable. The ratio of $\lambda_1:\lambda_2$ (i.e., the ratio of first constrained DCCA axis
1 and second unconstrained DCA axis 2) was used to assess the influence an explanatory variable has in
describing the variance in the chironomid community assemblage, and hence its predictive power (Juggins,
2013). All taxa were retained in the statistical analysis and rare taxa were down-weighted in the weighted
average transfer function (down-weighting of rare species is implicit in the Bayesian approach). Multivariate
analysis was carried out on square root transformed chironomid percentage data.



### 3.4 Inference models

Inference models were developed using two separate approaches. The first method relied on weighted averaging methods, a tried and tested technique well established in quantitative palaeoecology (Birks, 1998; Birks *et al.*, 2012; ter Braak and Juggins, 1993; ter Braak and Looman, 1986). The second method uses a Bayesian approach, which in general has received less attention (Holden *et al.*, 2008). There are a number of inherent problems associated with quantitative inference models (Huntley, 2012; Juggins, 2013; Velle et al., 2010) so the two independent methods were used to compare results and assess the strengths and weaknesses of each method.

### 3.5 Weighted averaging

The assemblage data was unimodal suggesting transfer functions using weighted averaging partial least squares (WA-PLS) were appropriate (ter Braak and Juggins, 1993). Inference models were also developed using classical and inverse weighted averaging (WA) to compare performance. The optimal number of components was assessed using leave-one-out cross validation (jack knifing) and a minimum 5% change in prediction error between components. Sample specific errors for the inferred temperatures were obtained through bootstrapping 999 cycles.

### 3.6 Bayesian

Bayesian model selection was used to generate probability-weighted species response curves (SRCs) for each taxon in the calibration dataset. Each taxon is assigned 8,000 possible SRCs. Each of these SRCs has a probability weight based on its relative ability to describe the training data for that taxon. To perform a reconstruction, likelihood functions (temperature probability distributions) are derived from each taxon in a fossil sample, considering all 8,000 SRCs. Combining the likelihood functions of all the taxa in the fossil sample derives the reconstruction. The power of the Bayesian approach is that it ascribes a probability distribution to the reconstruction, providing a reconstruction-specific uncertainty. An important benefit is that all taxa in the sample provide potentially useful information, even those with low counts that would be largely neglected in a



weighted averaging approach. To illustrate, a few counts of a taxon with a narrow temperature tolerance may
constrain the Bayesian reconstruction more than a very high count of a taxon with a broad tolerance.
Although the Bayesian model was developed for application to pH reconstructions from diatom
assemblages, it is generally applicable whenever it is appropriate to assume a unimodal species response to an
environmental gradient. The only modification required is the specification of appropriate priors. The *a priori*
probability distribution for optimum temperature in the SRCs was assigned to be uniform in the range -4.2 to
+30.8°C (training set range ±°5C). The *a priori* probability for SRC tolerance was assigned to be uniform in the
range 2 to 10°C. Other SRC priors were unchanged from those in Holden *et al.* (2008).

**3.7 Reliability of reconstructions**
DCCA detrending by segments, non-linear rescaling, and constrained by radiocarbon age was used to
determine compositional turnover constrained within the stratigraphic sequence (Birks and Birks, 2008). The
goodness-of-fit to temperature was evaluated by including the fossil chironomid samples passively in a CCA
ordination space of the modern training set samples constrained by MAT. Fossil samples with a squared residual
distance within the extreme 10% of the modern calibration dataset samples are considered as having a poor fit
to temperature. The modern analogue technique was used to test if fossil samples had good analogues within
the modern calibration data set. Any fossil sample with a squared chord distance larger than the 95% threshold
of the calibration data set is considered to have no good modern analogues (Birks, 1998; Velle *et al.*, 2005).
Data were untransformed prior to analysing the dissimilarity using the modern analogue technique. The
significance of the final reconstruction was tested by comparing the amount of variance in the fossil data
explained by that reconstruction, compared with inferences produced by transfer functions trained on
randomly generated environmental data (Telford and Birks, 2011a). In this case, 999 random environmental
variables were generated in order to produce the null distribution.





## 4. Results

### 4.1 Explanatory variables

The eight remaining explanatory variables, after those with VIFs >20 were removed; together explain 34.03% of the variance (Fig 2). The first two CCA axes explained 61.7% of the variance ($\lambda_1$=0.792, $\lambda_2$ = 0.466). MAT describes most of the variance in the chironomid assemblages and has the highest $\lambda_1$:$\lambda_2$ ratio (Table 2). When used as a single explanatory variable, MAT explains 12.93% of the variance ($\lambda_1/\lambda_2$= 1.431).

### 4.2 Calibration data set taxa

In total, 55 chironomid taxa were identified in the 59 training set lakes (Matthews-Bird et al., 2015). *Chironomus anthracinus*-type was the most widespread taxon, occurring over the entire temperature gradient (Fig 3). Orthocladiinae are generally most abundant towards the cold end of the temperature gradient (Matthews-Bird et al., 2015). *Cricotopus/Paratrichocladius* type III is the dominant taxon of the coldest lake and is not present in sites >10°C MAT. Figure 4 shows the weighted average and Bayesian optima and tolerance of each taxon ordered by lowest to highest optima as modelled in the weighted averaging approach. In general the temperature optima predicted by each method are similar, however, *Tanytarsus* type II and *Cricotopus/Paratrichocladius* type VII have colder optima when modelled using a Bayesian approach. *Cricotopus/Paratrichocladius* type IV has the coldest temperature optima *c.* 3.3°C (Fig 4). Few Chironominae were found at the cold end of the calibration data set, but, for example, *Parachironomus* and *Tanytarsus* type II were only found in lakes cooler than *c.* 8°C and had optima of *c.* 7.5°C and *c.* 6.5°C respectively. *Paratanytarsus* and *Pseudosmittia* are important components of the chironomid assemblage between 4-12°C, forming >50% of the chironomid community in some lakes, and have optima of *c.* 9.1 and 8.3°C respectively. *Tanytarsus* type I, *Micropsectra* and *Einfeldia* are dominant taxa at mid-temperatures between *c.* 10-22°C. The absence of lakes between *c.* 16°C and *c.* 20°C limits a complete understanding of the distribution of taxa occurring at these temperatures.

DCCA analysis, constrained by MAT, indicates an assemblage shift across the temperature gradient of 2.2 SD units. The biggest change in assemblage composition occurs above 12°C MAT (Fig 3). *Goeldichironomus*, *Cladotanytarsus* and *Tanytarsus* type III were only found in lakes with MAT warmer than *c.* 22°C. Tanypodinae



were in greatest abundance at the warm end of the temperature gradient between *c.* 10-26°C, *Procladius* was
the most common Tanypodinae. It occurred between *c.* 10-26°C and had an optimum of *c.* 21°C.
**4.3 Inference models**
Chironomid larval head capsule concentrations can vary significantly between lakes, due to differences in
preservation or abundance. Low counts can have adverse effects on the performance of inference models and
the reliability of quantitative environmental reconstructions when using conventional methods (Heiri and
Lotter, 2001; Quinlan and Smol, 2001). A minimum count size of 50 head capsules per sample is advised (Heiri
and Lotter, 2001; Quinlan and Smol, 2001), however, good model performance has been achieved even when
several samples include as few as 15-30 head capsules (Massaferro *et al.*, 2014). In some lakes in the current
training set head capsule concentrations were as low as two head capsules per gram of sediment. Fifteen lakes
in the data set produced fewer than 50 head capsules, and three lakes had fewer than 30. On average 77
individuals were analysed from each lake with a minimum count of 23 and a maximum of 164 (Table 1). Lakes
with low head capsule counts were retained in the model in order to maintain as even coverage as possible
across the temperature gradient.
Both methods (WA and Bayesian) produced similar performance statistics. The best performing model using
conventional statistical methods was a WA (inverse) model (Table 3, Fig 5)($R^2_{jack}$= 0.890, $RMSEP_{jack}$ = 2.404, Mean
bias$_{jack}$= -0.017, Max bias $_{jack}$=4.665). The Bayesian method produced a slightly higher performing model with
$R^2_{jack}$= 0.909, $RMSEP_{jack}$ =2.373, Mean bias$_{jack}$= 0.598, Max bias $_{jack}$= 3.158.


**4.4 Laguna Pindo fossil chironomids and dating**
Chironomid remains were found only in the upper 416 cm of the 929cm sequence of Laguna Pindo (Fig 6).
In total, 2489 individual chironomid head capsules were analysed. The entire assemblage was made up of 32
taxa in 26 genera and 4 subfamilies. Among the taxa identified, 17 were Chironomini, eight Orthocladiinae and
three Tanypodinae. There was high variation between samples both in number of head capsules (average: 82;
range: 24 - 184) and concentration per gram of wet sediment (average: 73; range: 2 - 163). There was a marked
decline in head capsule concentration below 200 cm. In younger sediments (200-0 cm) head capsule



concentration averaged 106/gram, in older samples (200-420 cm) the average was 44/gram. Five zones were
identified using optimal partitioning with a broken stick model to define significant zones. *Polypedilum nubifer*-
type, *Procladius* and *Limnophyes* were the most abundant taxa; abundances are over 10% wherever they
occurred. *Tanytarsus* type II was most abundant below 200 cm (1500 cal yr BP) whilst *Polypedilum nubifer*-type
was present in low numbers below 340 cm (2300 cal yr BP). During periods of low *Polypedilum nubifer*-type
abundance, *Tanytarsus* type II and *Tanytarsus* type I occur in greater numbers (e.g. 420-360; 290-250 cm).
The best-fit age depth model for Laguna Pindo was a smooth spline (Fig S1). Due to the absence of
chironomids at the bottom of the sequence, six radiocarbon samples were used for building the model with a
total depth of the sediment considered of 461 cm (Table S1). The sedimentation rate ranged between 0.03 and
0.5 cm/yr, with a sampling interval resolution of 97 years between samples on average (range from 27 to 196
years).
**4.5 Palaeotemperature reconstruction**
Both transfer functions (WA inverse and Bayesian) show similar patterns in the temperature
reconstruction (Fig 7). From 3000-2500 cal yr BP inferred temperatures are cold relative to the modern (20.2°C).
The minimum WA inverse temperatures are much colder (13.5°C±2.5) than the inferred Bayesian temperatures
(17.5°C±3.7) for the early section of the sequence. From 2400 to 1700 cal yr BP inferred temperatures from
both methods oscillate around *c.* 18-19°C but remained depressed relative to the modern. A notable feature of
both reconstructions is the sudden drop in inferred temperatures at 1600 cal yr BP. Inferred temperatures fall
by *c.* 2°C to 17.5°C±2.7. This abrupt drop in temperature is short-lived in both reconstructions and temperatures
return to previous values in the subsequent sample. From 1500 cal yr BP to the present the chironomid-inferred
temperatures stabilise and steadily rise. Peak temperatures for the entire record (21.9°C±3.5) are inferred
between 400-700 cal yr BP. Temperatures begin to cool from 400 cal yrs BP in both reconstructions, reaching a
minimum of *c.* 17°C±2.5 *c.* 100 cal yr BP before rising rapidly to between 20-21°C±2.5 in the most recent
sediment sample. On average the Bayesian model infers warmer temperatures than the WA model.
The fossil samples of Laguna Pindo plot within the modern variation of chironomid assemblages when
included passively in a CCA analysis of the calibration data set (Fig 8). This suggests that the calibration dataset
is appropriate for the fossil sequence of Laguna Pindo. The fossil samples plot along the MAP gradient



suggesting precipitation is an important variable controlling the variance in the fossil assemblages. The sites
associated with high precipitation in the calibration dataset are located in the same region of the Ecuadorian
Andes as the fossil site. With a modern MAT of *c.* 20°C, Laguna Pindo is located in a region of the temperature
gradient that is poorly covered in the calibration dataset (Fig 3), although the samples plot within the range of
modern calibration lakes that lie at similar elevations (1000-3000 m a.s.l). Seven taxa found in the Laguna Pindo
sequence do not occur in any of the analysed calibration data set lakes. These include three unknown
morphotypes, three *Xestochironomus* morphotypes, and *Metriocnemus eurynotus*-type. These taxa, however,
never comprise more than 10% of the chironomid assemblage of any one sample.
Seven of the fossil samples are considered to have a poor goodness-of-fit to temperature and all fossil
samples are considered as having poor modern analogues in the calibration data set (Fig 9). Although the
modern analogue technique is not used to infer past temperatures the lack of modern analogues in the fossil
assemblage is important when considering the reliability of any reconstruction.
DCCA constrained by radiocarbon age shows an abrupt change at 1475 cal yr BP between zones 3 and 4
and a turnover of 1.6 SD units over the whole sequence (Fig 9). The most recent sample is clearly distinct from
any other period of the record. Much of the variation in goodness-of-fit and DCCA sample scores is mirrored by
changes in count size and head capsule concentration. The sudden drop in head capsule concentration occurs at
a step change in DCCA assemblage variation (1475 cal yrs BP) (Fig 9). Periods of increased count size and head
capsule concentration in older sediments (2100-2250 cal yrs BP) also coincides with periods of improved
goodness-of-fit (Fig 9). The WA classical inferred MAT values using the modern calibration data set explain more
of the variance than 95% of randomly generated variables and so the WA classical MAT reconstructions can be
deemed statistically significant ($p$= 0.032) (Fig 10) (Telford and Birks, 2011a).

## 5. Discussion

### 5.1 Chironomids and environmental variables

Chironomids have been shown to respond to temperature at a variety of spatial scales and taxonomic levels
(Brooks, 2006; Eggermont and Heiri, 2011). Temperature is a key variable in controlling chironomid
development at all stages of their life cycles, and influences voltinism, behaviour and metabolism (Armitage *et*



*al*., 1995). Across the Northern Hemisphere, over large temperature gradients, mean July air temperature, the
warmest month of the year, which reflects the developmental period of most species, has been shown to be the
major determinant of variation in chironomid assemblages (Brooks, 2006; Walker and Cwynar, 2006). As a
result, many quantitative temperature inference models have been developed to reconstruct mean July air
temperature. Across the tropics however seasonal variation is small and many chironomids are multivoltine
(Walker and Mathews, 1987) so temperatures throughout the year are likely to be relatively more influential. In
tropical East Africa, Eggermont *et al*. (2010) demonstrated that mean annual air temperature was a significant
driver of chironomid assemblage composition and developed a chironomid-based inference model on this basis.
Similarly, Wu *et al*. (2014) showed MAT to be the most important environmental variable when developing a
chironomid inference model for Central America. When attempting to make quantitative inferences from fossil
assemblages it is first crucial to establish that the variable of interest is an important ecological determinant.
The variable to be reconstructed must describe a statistically important component of the variance within the
assemblage data (Juggins, 2013). Compared to other measured variables, mean annual temperature explained
the largest amount of chironomid assemblage variance and had the highest eigenvalue ratio ($\lambda_1:\lambda_2$) in the
Andean calibration dataset (Table 2). The explanatory strength of temperature in the calibration data set meets
the minimum criterion proposed by Juggins (2013) (i.e. $\lambda_1:\lambda_2 > 1.0$) for temperature being a suitable variable to
reconstruct from this calibration dataset.

The DCCA results suggest that precipitation is also a strong ecological determinant ($\lambda_1:\lambda_2=0.9$); the

passive plot of fossil samples with calibration samples further supports this conclusion. The fossil samples of
Laguna Pindo are strongly associated with MAP. Precipitation in Andean landscapes, however, is spatially
heterogeneous and geographically close localities experience significantly different rainfall patterns (Garreaud
*et al*., 2009). Lakes associated with high rainfall (Fig 2) are actually in areas of the northern Andes with two rainy
seasons a year. It is very likely that the bimodality of rainfall in these areas is as important in controlling
chironomid populations as the total amount of rainfall as measured by MAP. Precipitation is also intrinsically
linked to temperature as both temperature and precipitation increase with decreasing latitude in tropical South
America (Garreaud *et al*., 2009). Unlike temperature, precipitation affects chironomids indirectly making any
quantitative inference difficult. Precipitation will alter a suite of environmental variables (e.g. pH, conductivity,



depth, substrate) making quantitative inferences of precipitation problematic. As chironomid life cycles are
strongly controlled by temperature and many tropical chironomid species tend to be multivoltine, we suggest
the most appropriate variable both ecologically and statistically to reconstruct using the Andean calibration
data sets is MAT.

The optima and temperature tolerances (Fig 4) of many taxa found in the current study are similar to

that noted in other Neotropical chironomid calibration datasets, further supporting the conclusion of
temperature being an important ecological determinant. For example, Wu *et al*. (2014) in Central America,
found taxa of the genera *Beardius*, *Labrundinia* and *Goeldichironomus* to have optima between 23-24°C whilst
*Limnophyes* and *Corynoneura* where more abundant at the colder end of the gradient with optima of 15°C and
18°C respectively. In the current dataset *Beardius, Labrundinia*, and *Goeldichironomus* all have optima between
23-24°C and *Limnophyes* and taxa of *Corynoneura* also have optima of 15°C and 19°C, respectively. *Limnophyes*
also has one of the broadest tolerances of all taxa in both calibration datasets suggesting the genus is probably
represented by many species (Matthews-Bird et al., 2015). More work is needed in order to refine chironomid
larval taxonomy in South America, however the current data suggest the potential for a larger calibration
dataset applicable to wider area incorporating the Northern Neotropics and Central America.

**5.2 Model performance**

Although both models (WA inverse and Bayesian) perform well (WA RMSEP= 2.4°C and Bayesian

RMSEP= 2.3°C), some of the best performing chironomid-based temperature inference models have prediction
errors closer to 1.0°C (Brooks and Birks, 2001; Heiri *et al*., 2011, 2007; Olander *et al*., 1999). The highest
performing chironomid inference models often have in excess of 100-150 calibration sites compared with just
59 in the current model and this may account for its reduced performance. Furthermore the lakes in the
calibration data set are not evenly distributed over the temperature gradient. The cold end of the gradient has a
higher number of lakes (34 cold, high elevation lakes) than at warm and intermediate temperatures (15 warm,
mid-low elevation lakes). Uneven sampling has been shown to lead to biases which may reduce RMSEP (Telford
and Birks, 2011b). Furthermore the over-representation of cold lakes in the current dataset may result in under-
estimation of the temperature optima of some taxa and, therefore, bias temperature estimates towards cold



values. In the Andean dataset, as analysis of residuals shows, temperatures around 10°C are often under-
estimated (Fig 5). Furthermore, the inferred temperatures of Laguna Pindo are on average cooler than the
modern day conditions.
The absence of lakes in part of the temperature gradient may limit the reliability of estimates of optima
and tolerances of taxa and also create 'edge effects' in the middle of the temperature range, in addition to
those that occur at the cold and warm end of the temperature gradient (Eggermont *et al*., 2010). Such problems
are inherent to WA models as predicted values are pulled towards the mean of the training set resulting in
under- and over-estimations of high and low values (ter Braak and Juggins 1993). However, despite having no
lakes between 16-20°C in the calibration data set, additional edge effects are not a feature of the current
inference model. The gap of c. 4°C does not appear to have compromised model performance, probably as the
interval is not significant and taxa have tolerances that span these temperatures.
*Polypedilum nubifer*-type and *Chironomus anthracinus*-type make up a large component of the
chironomid assemblages in lakes across the entire temperature gradient (Fig 3). Such eurythermic taxa probably
include several different species. It is difficult to model reliable, or even meaningful, optima for eurythermic
taxa. Poor model performance or unreliable reconstructions may result if the assemblage is dominated by
eurythermic taxa. We note that eurythermic taxa are described by high tolerance SRCs in the Bayesian
approach, leading to increased uncertainty in reconstructions through broad likelihood functions that
contribute little information to the posterior. Inferred temperature of *c.* 10°C, are likely to be underestimated as
many taxa found at these temperatures also occur in cold lakes, which are over-represented in the calibration
data-set. In African lakes Eggermont *et al*. (2010) found that the presence of eurythermic taxa such as
*Chironomus* type Kibos caused an overestimation of temperatures in lakes at the warm end of the gradient.
They also found that the occurrence of *Limnophyes minimus*-type and *Paraphaenocladius* type OI Bolossat
overestimated the temperature of lakes close to where gaps occurred in the gradient (Eggermont *et al*., 2010).
Similarly, in a New Zealand calibration data set developed by Woodward and Shulmeister (2006), *Chironomus*
was present in both high elevation, cold, oligotrophic lakes and lower elevation, warm, eutrophic lakes. The
intermediate temperature optimum estimated for this taxon resulted in over-estimated temperatures of cold
lakes and under-estimates of warm lakes (Woodward and Shulmeister, 2006). Eurythermic taxa may be



contributing to the over-estimation of cold temperatures and the under-estimation of temperatures in the
middle of the gradient in the Andean inference model.

**5.3 WA vs Bayesian**
Despite similar performance statistics between the Bayesian and WA methods, the inferred pattern of late-
Holocene temperature change is different. Temperatures inferred c. 2700 cal yr BP (400 cm) (Fig 7) using the
WA inverse method is extremely cold (*c.* 14°C) compared with the rest of the record. This reconstruction is
driven by the high abundance of *Tanytarsus* type II, a taxon that has a WA temperature optimum of 6.5°C. The
Bayesian reconstruction for this sample of 17.8 ±2.8°C, is in line with more modest temperature shifts that
would be expected in the late-Holocene (Wanner *et al.*, 2008). One advantage of the Bayesian methodology is
the transparency of the reconstruction through consideration of individual likelihood functions for this
assemblage (Fig 11). Although *Tanytarsus* type II is abundant in the sample its influence in the reconstruction is
moderated by several other taxa with higher temperature optima that are present at low abundances. This
temperature estimate demonstrates the Bayesian reconstruction can be sensitive to a few counts of a species
that have a negligible effect in a WA approach. The likelihood function for Chironomini type II, which has an
abundance of only 2.3% in the sample, constrains the reconstruction more than *Tanytarsus* type II, which has an
abundance of 74%. This is because Chironomini type II is only found in the warmest lakes in the calibration set, each
time with a low abundance. We note that because it is found in only three training set sites, Chironomini type II is
associated with many (671) high-probability SRCs, defined as having a probability great that 10% of the most likely
SRC. For this reason, its likelihood function is relatively broad and extends to temperatures far lower than the
temperature of the sites in which the taxon is found in the training set.

**5.4 Laguna Pindo temperature reconstruction**
It is important that the study site is appropriate for the calibration data set. The variable of interest must be
the most important in driving biotic change (Velle *et al.*, 2005) especially considering the influence of important
secondary variables. It would be difficult to generate statistically significant reconstructions from fossil
assemblages that are influenced by several variables acting together (Telford and Birks, 2011a). Laguna Pindo is



located at 1248 m a.s.l on the eastern Andean flank at the transition between the cold environments of the high
Andes and warmer lowlands of the Amazon basin. Therefore, cold climatic periods during the lake's history
would promote the migration of cold stenothermal taxa from higher elevations down the Andean flank to
occupy the lake (Čiamporová-Zaťovičová *et al.*, 2010). Conversely, during warm periods, taxa inhabiting lowland
lakes would move up the Andean flank. Despite the influence of precipitation the location of Laguna Pindo
makes it a good palaeoecological setting to record the response of temperature-sensitive proxies.

**5.5 Temperature and secondary environmental variables**
Whilst the $\lambda_1/\lambda_2$ of 1.431 indicates that MAT is appropriate for reconstruction using this calibration dataset
(Juggins, 2013), it does not necessarily mean that reliable temperature reconstructions can be obtained from a
fossil record (Telford and Birks, 2011a). Before attempting to interpret any reconstruction several metrics can
be used to assess the validity of a reconstruction (Juggins and Telford, 2012).
The modern analogue technique compares the similarity of the fossil samples to the modern samples in the
calibration data set. All fossil samples are greater than the 5[th] percentile of the square chord distance (Fig 9),
which suggests there is no close modern analogue in the calibration set to any fossil sample (Birks, 1998; Juggins
and Birks, 2001). The lack of modern analogues in the Laguna Pindo fossil sequence is due to the many taxa
present in the fossil samples that are not present in the calibration data set. This may reflect the lack of lakes in
the calibration dataset with MAT values close to those of Laguna Pindo. Nevertheless, WA and WAPLS models
have been shown to perform well in non-analogue situations (Birks *et al.*, 2010). The Bayesian method
generates temperature reconstructions from likelihood functions of species in the calibration data set. Although
analogous assemblages are not required for the Bayesian reconstruction (each taxon is treated equally and
individually), species that are absent from the training set cannot contribute information to the posterior,
thereby increasing the uncertainty associated with the reconstruction. One advantage of the Bayesian
methodology is that this uncertainty is explicitly incorporated into the Bayesian reconstruction (Holden *et al.*,

2008).

During periods of poor fit-to-temperature, variables other than temperature may have been affecting the
composition of the chironomid assemblage. As noted previously, the CCA biplot of fossil samples included



passively with the significant explanatory variables (Fig 8) shows that MAP was also important in driving the
assemblage variance. During times of poor fit to temperature the influence of precipitation as a secondary
variable may be more important than temperature in influencing the chironomid assemblage composition.
Indeed, precipitation has been shown to be an important variable in controlling the modern distribution of
chironomid taxa in the tropical Andes (Matthews-Bird et al., 2015).

Samples with poor fit-to-temperature also corresponded with samples having low numbers of head

capsules. The number of head capsules retrieved will directly affect how representative a sample is to the
chironomid fauna (Heiri, 2004; Quinlan and Smol, 2001). The cold oscillations inferred from the Bayesian
reconstruction are more in line with what is expected during the late-Holocene (1-3°C); the likelihood functions
of rare species which favour warm conditions combine to rule out the anomalously cold temperatures
suggested by some of the WA reconstructions. As discussed above, the over-representation of cold lakes in the
calibration dataset will likely bias species optima to colder values in a weighted average approach so there may
be a tendency for the model to underestimate temperature, especially during cold periods. This problem is
likely exaggerated when head capsule concentration is low, cold indicator taxa may have higher abundances
than would be the case if all taxa were accurately represented.

The DCCA results indicate that there was a distinct change in the composition of the chironomid

assemblage after 1600 cal yr BP (210 cm). This largely coincides with an increase in head capsule concentration,
possibly indicating an increase in lake productivity, and the shift in chironomid-inferred temperatures from low
to high. Indeed post 1600 cal yr BP, (210 cm) samples are inferred as being on average 2-3°C warmer than early
sections using Bayesian and WA models respectively.

Although the temperature reconstruction has a good ecological basis, because chironomids globally are

highly sensitive to temperature and Laguna Pindo is on an ecotonal boundary that is sensitive to temperature
changes, precipitation is influential as a secondary variable. The WA inverse MAT reconstruction, however, is
statistically significant based on the criteria described by Telford and Birks (2011a) (Fig 10) suggesting that
despite conflicting variables a temperature signal can be obtained from Neotropical chironomids.



### 5.6 Cooling climate 3800-2800 cal yrs BP

Changes in insolation are often inferred as the dominant driver of Holocene climate changes (Mayewski *et al.*, 2004). The period between 3500-2500 cal yr BP is recognised globally as a period of rapid climate change, with changes in precipitation and temperature recorded globally from a range of different proxies (Mayewski *et al.*, 2004; van Geel *et al.*, 1999). The most notable feature of South American Holocene climate is the migration of the ITCZ, which affects rainfall patterns and is caused by changes in insolation affecting tropical Pacific sea surface temperatures (Bird *et al.*, 2011; Haug *et al.*, 2001). The period between 3800-2800 cal yrs BP is a time of erratic rainfall in the Caricao Basin of Venezuela, which resulted in pronounced aridity (Haug *et al.*, 2001) and coincides with a maximum in $\Delta^{14}$C and $^{10}$Be, indicating a decline in solar output (Mayewski *et al.*, 2004). The low chironomid-inferred temperatures between 3000-2500 cal yr BP probably reflect this decline in solar output allowing high Andean taxa to migrate down slope. Our data suggests reduced insolation, as well as changing precipitation patterns, could have the effect of cooling tropical climate.

### 5.7 Recent cooling

Prominent among climate fluctuations of the last millennium is the Little Ice Age (LIA), an event recognised globally from historical and proxy climate records (Jones and Mann, 2004; Mayewski *et al.*, 2004). The peak and duration of the LIA varies around the globe. In the tropical Andes a cold period, identified as the LIA, has been recorded as early as 1180 AD (770 cal yr BP) and lasting as late as 1820 AD (130 cal yr BP) based on evidence of glacial advance in mountain glaciers (Polissar *et al.*, 2006). The extent of LIA fluctuations remain unclear in the tropics due to the paucity of quantitative palaeoclimate data (Crowley, 2000). Variation in the sun's energy output is regarded as the main driver of LIA cooling. Although the tropics receive 47% of planetary insolation, the climate response in the tropics to solar variability is poorly understood (Crowley, 2000; Polissar *et al.*, 2006). The chironomid-inferred temperatures from Laguna Pindo suggest cooler temperatures from 400 cal yrs BP (50 cm) (1550 AD), reaching a minimum *c.* 100 cal yrs BP (20cm)(1850 AD), before rising again to the present day. The inferred temperature minimum for this period is 17.7 ± 3.8°C which suggests a cooling of 2.2°C relative to the modern temperature. We caution this estimate is within the RMSEP of the inference model and only recorded in a single sample making any interpretations tentative. Multiple lines of evidence, however, do



support a cooling in the tropical Andes similar in magnitude to that inferred here. Glacier fluctuations indicate a
cooler and wetter climate than today for the outer and inner tropics of South America between the 17[th] and
19[th] century (Jomelli *et al.*, 2009). The Huascaràn and Quelccaya ice core records (Thompson *et al.*, 1986;
Thompson *et al.*, 1995) from Peru suggest the latter half of the Holocene was a period of long-term cooling,
reaching a minimum during the LIA, synchronous with Northern Hemisphere cooling. Temperature estimates
from stable isotope data are difficult to generate, however, as the proxy is affected by changes in snow surface
histories (Thompson *et al.*, 1986; Thompson *et al.*, 1995). Polissar *et al* (2006), using pollen and evidence of
glacial retreat from the Venezuelan Andes, estimated a mean annual temperature reduction of 3.2±1.4°C
relative to modern averages during the LIA. The chironomid-inferred temperatures at Laguna Pindo do not
show the late-Holocene cooling trend culminating in the LIA, as recorded in the Andean ice core records.
Instead temperatures remain relatively stable from 1500 cal yrs BP (195 cm), with a peak for the entire 3000
year record c. 400 cal yrs BP (50 cm) (21.8°C), followed by a drop in temperatures of 3-4°C beginning around
200 cal yrs BP (30 cm). This fluctuation is similar in magnitude to other records from the tropical Andes (Polissar
*et al.*, 2006) however a higher resolution dataset is needed to investigate this trend further.
**6.  Conclusions**
The chironomid fauna of the tropical Andes have been shown to be sensitive to climate variables,
particularly temperature and precipitation. Both variables (MAT and MAP) meet the basic criteria for being used
in an environmental reconstruction using the Andean calibration dataset. MAT, however, is an important
determinant of chironomid species distribution and abundance and was therefore more appropriate to be
reconstructed. The influence of precipitation should be explored further and must be considered as an
important secondary variable especially when reconstructing past conditions in the region. It is very likely that
the influence of precipitation noted here relates to the annual variability in rainfall across the Andes as opposed
to overall amount making any quantitative interpretations even more difficult.
The two techniques used to develop inference models (WA and Bayesian) show comparable performance
statistics (WA inverse model $R^2_{jack}$= 0.890, $RMSEP_{jack}$ = 2.404, Mean bias$_{jack}$= -0.017, Max bias $_{jack}$=4.665; Bayesian
model $R^2_{jack}$ 0.909, $RMSEP_{jack}$ =2.373, Mean bias$_{jack}$= 0.598, Max bias $_{jack}$= 3.158). This work demonstrates a





proof of method, however, a larger calibration dataset with a more even coverage of calibration sites is needed
in order to improve model performance. The Bayesian approach provided a transparent reconstruction less
susceptible to the effect of an uneven distribution of calibration sites and performed particularly well during
periods of low count size and when inferring cold intervals. The chironomid-based MAT reconstruction from the
Laguna Pindo fossil record suggests that periods of low solar output not only affect the tropics through changes
in precipitation, but also directly affect tropical temperatures. Inferred temperatures were 2-3°C cooler relative
to the modern during the widely recognised 3500-2500 cal yr BP cooling event. Long-term cooling during the
late Holocene is not apparent in the Laguna Pindo record. However, temperatures do cool by 1-2.2°C relative to
the modern during the LIA period, although this is only noted in a single fossil sample.

Knowledge of past tropical climate dynamics is fundamental not only to understanding regional climate

but also global climate patterns and hemispherical teleconnections. Quantitative temperature proxies, such as
chironomids, will provide valuable data on past climate variability in the region. The reconstructions presented
here demonstrate the potential of the proxy and also highlights the complexity of late-Holocene climate change
in tropical South America.


**Acknowledgements**

Funding was provided by the Natural Environment Research Council (NERC), UK. NERC grant (ref:
NE/J018562/1) was awarded to E. Montoya and (ref: NE/J500288/1) awarded to F. Matthews-Bird. This work
was supported by the NERC Radiocarbon Facility NRCF010001 (allocation number 1682.1112) Special thanks to
Dr Pauline Gulliver for her continuous involvement and support during radiocarbon dating. The authors also
wish to thank Mark Bush, Francis Mayle, Yarrow Axford, Alex Chepstow-Lusty and Mick Frogley for their kind
donation of samples.




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

**Table captions**
**Table 1**
Summary of the physical and chemical properties of the 59 calibration data set lakes including the total number
of head capsules retrieved from each lake and the concentration of head capsules per gram of sediment.  MAT=
mean annual temperature, MAP= mean annual precipitation, LOI=loss-on-ignition.
**Table 2**
Results of detrended canonical correspondence analysis (DCCA) using single constraining variables. MAT= mean
annual temperature, WT=water temperature, MAP=mean annual precipitation, LOI= Loss-on-ignition.
**Table 3**
Summary of the performance statistics of chironomid-based MAT inference models developed using classical
and Bayesian methods based on leave one out cross validation. Weighted averaging inverse and classical
(WAinv, WAcla), Weighted averaging partial least squares (WA-PLS), coefficient of determinant between
predicted and observed ($r^2_{jack}$), root mean squared error of prediction ($RMSEP_{jack}$) as % of the gradient.


**Tables**
**Table 1**

| | Calibration data set | | | | |
|---|---|---|---|---|---|
| | Minimum | Mean | Median | Maximum | Std dev |
| **Conductivity (μs)** | 5.9 | 363 | 185 | 3205 | 579 |
| **Depth (m)** | 0.1 | 5 | 2.2 | 25 | 5.4 |
| **Elevation (m a.s.l)** | 150 | 3142 | 3845 | 4655 | 1459 |
| **Latitude (S)** | 0.1 | 11.2 | 14.2 | 17.3 | 6.2 |
| **Longitude (W)** | 64.4 | 71.6 | 70.3 | 78.4 | 4.5 |
| **LOI  (%)** | 0 | 19 | 13 | 80 | 16 |
| **MAT (°C)** | 0.8 | 12 | 10 | 25 | 7 |
| **MAP (mm/year)** | 468 | 1222 | 769 | 4421 | 952 |
| **pH** | 5.7 | 8 | 7.9 | 10.2 | 1.1 |
| **Total Head Capsules** | 23 | 77 | 76 | 164 | 35 |
| **Water Temperature (°C)** | 5 | 15 | 13 | 33 | 6 |
| **Head capsule/gram** | 2 | 27 | 22 | 105 | 22 |


**Table 2**

| Variable | Variance Explained (%) | $\lambda_1/\lambda_2$ | $P$ |
|---|---|---|---|
| **MAT** | 12.93 | 1.431 | 0.001 |
| **MAP** | 10.3 | 0.900 | 0.001 |
| **WT** | 11.21 | 1.230 | 0.001 |
| **pH** | 6.23 | 0.500 | 0.001 |
| **LOI** | 3.23 | 0.239 | 0.062 |
| **Depth** | 2.44 | 0.190 | 0.240 |
| **Conductivity** | 2.34 | 0.179 | 0.296 |




**Table 3**

| Model | $R^2_{Jack}$ | $RMSEP_{jack}$ | Mean bias$_{jack}$ | Max bias$_{jack}$ | % change |
|---|---|---|---|---|---|
| WA (inv) | 0.890 | 2.404 | -0.017 | 4.665 | - |
| WA (cla) | 0.890 | 2.475 | -0.035 | 4.279 | -2.936 |
| WA-TOL (inv) | 0.851 | 2.831 | -0.182 | 6.498 | - |
| WA-TOL (cla) | 0.852 | 2.951 | -0.211 | 7.350 | -4.263 |
| WA-PLS (1) | 0.889 | 2.431 | 0.094 | 4.891 | - |
| WA-PLS (2) | 0.890 | 2.412 | 0.109 | 3.982 | 0.766 |
| WA-PLS (3) | 0.869 | 2.617 | 0.096 | 5.558 | -8.483 |
| WA-PLS (4) | 0.866 | 2.659 | 0.199 | 5.922 | -1.592 |
| WA-PLS (5) | 0.875 | 2.568 | 0.213 | 6.201 | 3.409 |
| Bayesian | 0.909 | 2.373 | 0.598 | 3.158 | |





**Figure captions**
**Figure 1**
Location of the calibration data set lakes (black circles) and Laguna Pindo (white triangle).
**Figure 2**
Figure 2: Canonical correspondence analysis (CCA) of the calibration data set lakes and environmental variables
with elevation and longitude removed after variance inflation analysis. MAP=mean annual precipitation,
MAT=mean annual temperature, WT= water temperature, LOI=loss-on-ignition. Grey circles denote calibration
lakes, dark grey triangles mark species. All species could not be labelled due to crowding; instead nine
important taxa have been marked as examples.
**Figure 3**
Chironomid taxa in the modern calibration dataset lakes. Lakes are ordered (top to bottom) from cold to warm
and chironomids are ordered by occurrence from cold to warm lakes. Only taxa present in three or more lakes
are included. Dashed line shows a gap in calibration data set lakes between 16-20 °C of the MAT gradient.
Detrended canonical correspondence analysis (DCCA) constrained by MAT shows the taxon turnover across the
gradient. Head capsule concentration (hc/gram) is also included.
**Figure 4**
Weighted-average and Bayesian optima (*solid grey circles*) and tolerances (*thick lines*) of the 55-chironomid taxa
included in the calibration dataset, MAT Range (*dashed lines*). Taxa are organised by WA temperature optima
from cold to warm.
**Figure 5**
Model performance of the best performing classical method (WA) and Bayesian approach. A=weighted
averaging method; B=Bayesian method. WA: $R^2_{jack}$= 0.890, $RMSEP_{jack}$ = 2.404, Mean $bias_{jack}$= -0.017, Max bias
$_{jack}$=4.665. Bayesian: $R^2_{jack}$= 0.909, $RMSEP_{jack}$ =2.373, Mean $bias_{jack}$= 0.598, Max bias $_{jack}$= 3.158.



**Figure 6**
Diagram of fossil chironomid assemblage of Laguna Pindo. Five significant zones were identified using optimal
partitioning with a broken stick model. Detrended canonical correspondence analysis (DCCA) constrained by
calibrated radiocarbon age shows taxon turnover through time.  Only taxa with relative abundances greater
than 5% are shown. SD=standard deviation, hc/gram= head capsules per gram of wet sediment.
**Figure 7**
Chironomid-inferred mean annual temperatures (MAT) at Laguna Pindo using the WA inverse (grey) and
Bayesian (black) models. Sample specific errors for the WA model are obtained through bootstrapping 999
cycles. Errors of the Bayesian reconstruction are site-specific uncertainties.  Key late-Holocene climate events
are shaded in grey. LIA=the range of the earliest and latest date for the Little Ice Age in South America (Polissar
*et al.*, 2006). 3500-2500 global cooling event (Mayewski *et al.*, 2004), note, however, the Laguna Pindo record
only extends to 3000 cal yrs BP.
**Figure 8**
Distribution of Laguna Pindo fossil samples (black circles) included passively within a CCA of the calibration data
set lakes (grey circles) constrained using the significant environmental variables. MAP= mean annual
precipitation, MAT= mean annual temperature, WT= water temperature. The first and last fossil sample in the
sedimentary sequence has been labelled (total sediment depth); there are no directional trends through time.
Calibration lakes that lie at similar elevations as Laguna Pindo have been labelled.


**Figure 9**
(left to right): Chironomid-inferred WA classical MAT with sample specific errors generated using bootstrapping.
Bayesian reconstruction with sample specific errors. Goodness-of-fit of the fossil assemblages to temperature,
vertical dotted line indicates the 90[th] percentile of squared residual distances of modern samples to first axis in
a CCA; samples to the right of the line have a poor fit-to-temperature. Nearest modern analogue analysis,





vertical dotted line indicates the 5[th] percentile of squared chord distances of the fossil samples in the modern
calibration data set; samples to the right of the line have no good modern analogues. Detrended canonical
correspondence analysis (DCCA) sample scores with radiocarbon age used as the sole constraining variable.
Head capsule concentration per gram of sediment. Zones are derived from optimal partitioning of fossil
assemblages using a broken stick model to define significant zones. Sq res dis= square residual distance; Sq chrd
dis= square chord distance; SD units= standard deviation units; hc/gram=head capsule per gram of sediment.
**Figure 10**
Histogram of the proportion of variance in the chironomid MAT transfer function explained by 999 transfer
functions trained on random environmental variables. Solid black line denotes the proportion of variance
explained by the chironomid WA inverse MAT transfer function. Black dashed line marks the proportion of
variance explained by the first axis of PCA of the fossil data. Grey dashed line marks the 95% variance of the
random reconstructions.
**Figure 11**
Individual likelihood functions for the fossil taxa in the coldest sample of the Laguna Pindo sequence (396 cm
total depth, *c.* 2700 cal yr BP). The posterior probability distribution for temperature for the fossil sample is
plotted in red, note this is plotted on an independent axis.











**Figures**

**Figure 1**






**Figure 2**


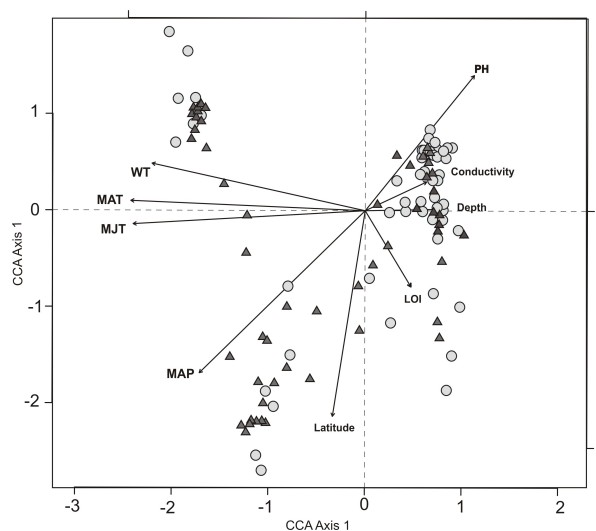



**Figure 3**





**Figure 4**



**Figure 5**





078    **Figure 6**



**Figure 7**

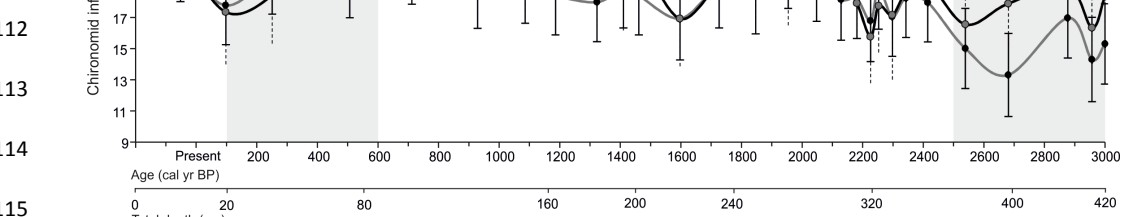





**Figure 8**

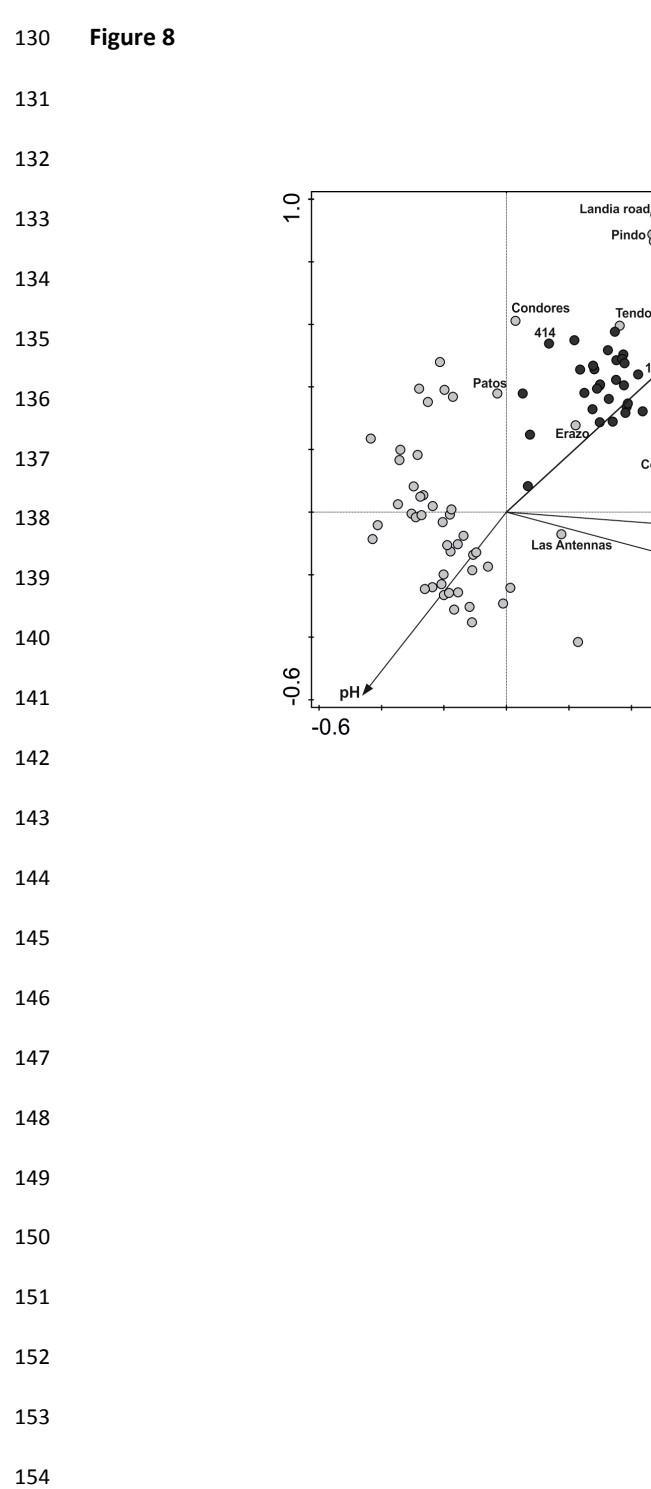





**Figure 9**







**Figure 10**

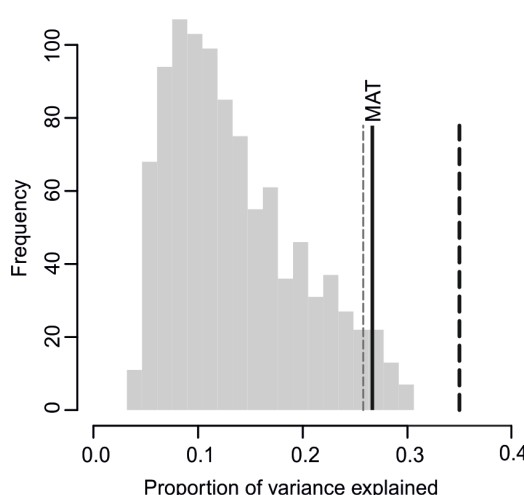





**Figure 11**



















