# Peer review of "Published: 20 January 2016 © Author(s) 2016. CC-BY 3.0 License."

_Climate of the Past, 2015_

## Short Comment (SC1) · 22 Jan 2016

Late Holocene reconstructions using multiple climate proxies are of greatest importance. The palaeoclimate results represent important calibration data for modern climate change and respective models. Under certain circumstances, chironomid data provide valuable palaeotemperature information. Pitfalls exist, as the authors have discussed in detail themselves (e.g. Brooks et al. 2012).

The reconstructed temperature curve is interesting. I am particularly looking at the last 1000 years. The number of samples is unfortunately rather small (every 100-200 years) and does not allow full resolution of the Medieval Warm Period and Little Ice Age. The interpolated curve segments may therefore look quite different when higher

resolution data was plotted. Maybe future research on the same material can infill data here?

There is a number of nearby studies which are worth comparing to the new curve. Most of them record precipitation, but there are also a few temperature reconstructions. The respective studies are mapped here: https://www.google.com/maps/d/viewer?mid=zvwgQ0tAjx_k.keO5eR4ueHXE Click on the dots to get key information including the main graph. Yellow indicates a dry MWP, red a warm MWP.

The closest study is Ledru et al. 2013 http://www.clim-past.net/9/307/2013/cp-9-307-2013.html The study reconstructs precipitation. Result: Dry phase 1250-1550 AD, characterized by an abrupt decrease in the T/P index. During the Little Ice Age, two phases were observed: first, a wet phase between 1550-1750 AD, followed by a cold dry phase 1750-1800 AD. How does this fit with the new results?

There is another study by Mayewski et al. 2004, Rodbell et al. 1999 in Laguna Pallcacocha that found a dry phase 700-1200 AD. Needs discussion.

The next temperature reconstructions in the region that I am aware is from Kellerhals et al. 2010 who write in their abstract: "For the time period from about 1050 to 1300 AD, our reconstruction shows relatively warm conditions that are followed by cooler conditions from the 15th to the 18th century, when temperatures dropped by up to 0.6°C below the 1961-1990 average. The last decades of the past millennium are characterized again by warm temperatures that seem to be unprecedented in the context of the last 1600 years."

How do the new temperatures fit with this (higher resolution, different proxy) reconstruction? Other interesting temperature papers are from Salvatteci et al. 2014 & 2016 and Zuluaga et al. 2015: Offshore core G10. Results of Salvatteci et al. 2016: Organic-rich interval 1000-1400 AD indicates warm conditions with strongly developed oxygen-minimum zone (OMZ). Weak OMZ with low organic content sediment during

subsequent cold phase of Little Ice Age.

It would be good if the authors could discuss their results in the light of nearby studies which would help to give more confidence in the validity of the technique and results.

---

## Author Comment (AC1) · 27 Jan 2016

We agree that palaeoclimate proxies are an invaluable component of climate science. The data generated by palaeoclimate proxies are important calibration points for modern and future climate science. Multi proxy studies from around the globe are vital for refining our understanding of long-term climate change and global teleconnections. Chironomidae, however, as a palaeoclimate proxy in tropical South America are still very much in their infancy. This study presents the first calibration dataset and the first quantitative temperature reconstruction using chironomids from the region.

As Dr Luening points out unfortunately the resolution of the fossil data is low and indeed the interpretation may change with a more detailed record. For this reason we Full screen / Esc

cautioned against an over interpretation of the record. Our comments on this matter are confined to two brief paragraphs in the conclusion. We merely wished to demonstrate a proof of method, a better understanding of the different models (WA and Bayesian), and some interesting trends in the fossil data, that as Dr Luening highlights, further research can infill.

Dr Luening provides a detailed outline of the palaeoclimate research around our site. We wish to thank him for this contribution, and for fostering some important discussion on the past climate of tropical South America. The link to the map of palaeoclimate reconstructions from the region will be particularly important to many researchers working in the area.

However, as Dr Luening highlights, many of these reconstructions are of past precipitation. Even the few reconstructions of past temperature, namely the oxygen isotopic records of the Andean ice cores, have a strong precipitation component. As a result, it is difficult to place direct temperature estimates on the observations from isotopic records. We confine the discussion of the temperature variability observed in our record to studies with direct temperature estimates. We acknowledge that this is by no means an exhaustive list of the palaeoclimate research in the region but it does provide the most meaningful comparison to this study whilst contributing to our overarching aiming, that of assessing the utility of chironomidae in the area.

Once more we wish to thank Dr Luening for his comment, all the points he raises are valuable contributions to this discussion. We will update the final manuscript with an expanded discussion including the records mentioned. These include the near by precipitation records of Laguna Pallcacocha (Mayewski et al. 2004; Rodbell et al. 1999) and the Sucus bog at Papallacta (Ledru et al. 2013), and the temperature reconstructions from the Bolivian Ice core of the Nevado Illimani (Kellerhals et al. 2010).

---

## Referee Comment (RC1) · D. Porinchu (Referee) · 19 Feb 2016

Quantifying late Holocene climate in the Ecuadorian Andes using a chronomid‐based temperature inference model Matthews-Bird et al. Climate of the Past

Summary The training set developed by Matthews-Bird et al., based on surface sediment recovered from 59 Andean lakes in tropical South America, is used to: 1) determine the distribution of subfossil midge remains; 2) identify environmental variables that can account for a statistically significant amount of variance in the distribution of the midges; and 3) produce a midge-based inference model for mean annual temperature (MAT). In addition, the inference model, which is applied to the subfossil midge assemblages preserved in a sediment core recovered from Laguna Pindo, a small, shallow lake in the eastern Andes of Ecuador, is used to develop a quantitative midge-based reconstruction of late Holocene temperature change for region.

The development of a midge training set and the associated inference model for this region is notable and important. This study provides much needed insight into the biogeography of midges in tropical South America and also serves to improve our understanding of the modern midge-environment relationship. I believe that this paper makes an important contribution and should be accepted for publication with some revisions. My major concerns are outlined below with minor issues following.

Major Concerns

1. The range in midge-inferred MAT values at Laguna Pindo during the late Holocene is $\sim$ 6oC. In addition, there are intervals where the change in midge-inferred MAT is of a large magnitude and quite rapid (e.g. the $\sim$ 5oC decrease in MAT that occurs between 400 and 100 cal yr BP). MAT is inherently less variable than seasonal or monthly temperature estimates; this is especially true in the very low latitudes where MAT does not vary appreciably through the year. I believe the authors should include a discussion of the potential drivers of tropical climate during the late Holocene given the large magnitude fluctuations in inferred temperature documented at Laguna Pindo. What is driving a 5oC change in MAT during the Little Ice Age (LIA)? Is the magnitude and rate of this change in inferred MAT reasonable? Polissar et al. (2006) is cited to suggest that the change in MAT at Laguna Pindo during the LIA is comparable to other records from the topical Andes; however the reconstructions in Polissar et al. (2006) are based on sites located at $\sim$ 4200 m asl ($\sim$ 3200 m higher in elevation than Laguna Pindo) and likely influenced by vertical amplification of warming in the tropics (Thompson et al. 2011).

2. Are there alternative explanations/site specific drivers that can account for the shifts in the midge assemblages at Laguna Pindo? For example, could changes in lake level and/or the composition of aquatic vegetation influence the midge assemblages and thereby, confound downcore interpretations and the midge-inferred temperature reconstruction.

3. A number of studies identify ENSO-related shifts climate in the eastern tropical Pacific (e.g. Conroy et al. 2008) and the western margin of tropical South America (e.g. Moy et al. 2002) during the late Holocene. What role does ENSO variability play in influencing the midge assemblages at Laguna Pindo? The Medieval Climate Anomaly (MCA) is also well expressed in the low latitudes (Mann et al. 2009); however, no mention of the MCA is made in the manuscript. Is the influence of MCA-related solar and volcanic forcing evident at the study site? The authors should more explicitly connect the downcore reconstruction to known drivers of late Holocene climate change in the region.

4. The following concerns relate to the robustness and reliability of the MAT reconstruction. The reliability if the reconstruction is assessed using a number of standard approaches including modern analogue technique and a goodness-of-fit measure. These approaches document that the downcore samples are not well represented in modern training set and many of the samples have a poor-fit to temperature. In addition, there appears to be a correspondence between samples with low head capsule counts and samples that have a poor fit to temperature. Lastly, there is a fairly large discrepancy between the midge-inferred estimate of modern MAT ($\sim$ 17oC) and the observed MAT for the study site (20.2oC). The above highlight some of the issues related to the quantitative reconstruction. A more detailed discussion on the paleoecological information captured by variations in midge assemblage composition would help to strengthen the paper. In addition, elaborating on the results of the DCCA would provide interesting qualitative information on the timing and magnitude of faunal turnover.

5. L. 542 : Is the primary control on the subfossil midge assemblages temperature or precipitation? The assertion that taxa associated with higher sites are migrating down slope in response to changing climate can be substantiated by passively plotting fossil assemblages against the modern training set samples (coded/classified by elevation)

in ordination space.

Minor Issues

1. l. 349: I do not think that that lakes located between 1000 and 300 m asl can be considered to be similar in elevation to Laguna Pindo ($\sim$ 1200 m asl). Applying a standard lapse rate to this elevation range suggests that MAT for the lowest and highest lakes would vary by 12-20oC.

2. l. 415: reporting the RMSEP as a % of the total MAT range captured by the training set would be useful.

3. Fig. 1: requires a N-arrow

4. Fig 2: "PH" should be corrected.

5. It is not clear why non-limnological variables such as latitude were included in the exploratory analysis. Latitude, longitude and elevation are not directly controlling the distribution of midges; the analyses should be re-run with only environmental variables that have the potential to directly control the distribution of midges included.

References Cited

Conroy, J.L., Restrepo, A., Overpeck, J.T., Steinitz-Kannan, M., Cole, J.E., Bush, M.B. and Colinvaux, P.A., 2009. Unprecedented recent warming of surface temperatures in the eastern tropical Pacific Ocean. Nature Geoscience, 2(1), pp.46-50.

Thompson, L.G., Mosley-Thompson, E., Davis, M.E. and Brecher, H.H., 2011. Tropical glaciers, recorders and indicators of climate change, are disappearing globally. Annals of Glaciology, 52(59), pp.23-34.

Mann, M.E., Zhang, Z., Rutherford, S., Bradley, R.S., Hughes, M.K., Shindell, D., Ammann, C., Faluvegi, G. and Ni, F., 2009. Global signatures and dynamical origins of the Little Ice Age and Medieval Climate Anomaly. Science, 326(5957), pp.1256-1260.

Moy, C.M., Seltzer, G.O., Rodbell, D.T. and Anderson, D.M., 2002. Variability of El Niño/Southern Oscillation activity at millennial timescales during the Holocene epoch. Nature, 420(6912), pp.162-165.

Please also note the supplement to this comment:
http://www.clim-past-discuss.net/cp-2015-186/cp-2015-186-RC1-supplement.pdf

———————————————————

[Figure]

**Supplement:**

[revised manuscript text omitted]

                                               **Figures**

**Figure 1**

[Figure]

[Figure]

**Figure 2**

[Figure]

[Figure]

[Figure]

**Figure 3**

[Figure]

[Figure]

[Figure]

**Figure 4**

[Figure]

[Figure]

[Figure]

**Figure 5**

[Figure]

[Figure]

**Figure 6**

[Figure]

[Figure]

**Figure 7**

[Figure]

[Figure]

[Figure]

**Figure 8**

[Figure]

[Figure]

[Figure]

**Figure 9**

[Figure]

[Figure]

[Figure]

**Figure 10**

[Figure]

[Figure]

[Figure]

**Figure 11**

---

## Referee Comment (RC2) · Anonymous Referee #2 · 22 Feb 2016

1. Does the paper address relevant scientific questions within the scope of CP? YES

2. Does the paper present novel concepts, ideas, tools, or data? YES

3. Are substantial conclusions reached? See comments. I don't' think that the Conclusions are sufficiently supported by the data (fit-to temperature, representation of modern analogues in fossil samples, suitability of L Pindo wrt to the training set, possibly also data resolution, etc.)

4. Are the scientific methods and assumptions valid and clearly outlined? (See comments below)

5. Are the results sufficient to support the interpretations and conclusions? Partly

6. Is the description of experiments and calculations sufficiently complete and precise to allow their reproduction by fellow scientists (traceability of results)? See comments below.

7. Do the authors give proper credit to related work and clearly indicate their own new/original contribution? Yes (see comments)

8. Does the title clearly reflect the contents of the paper? No

9. Does the abstract provide a concise and complete summary? YES

10. Is the overall presentation well structured and clear? YES (few minor comments)

11. Is the language fluent and precise? YES

12. Are mathematical formulae, symbols, abbreviations, and units correctly defined and used? Generally YES

13. Should any parts of the paper (text, formulae, figures, tables) be clarified, reduced, combined, or eliminated? NO

14. Are the number and quality of references appropriate? YES (see comments)

15. Is the amount and quality of supplementary material appropriate? I would include the chronology in the manuscript (NOT in SOM)

General remarks: This manuscript presents, to my knowledge, the first chironomid-based Transfer Function in tropical South America from >50 lakes in Peru and Ecuador, and provides a temperature reconstruction from fossil samples of Laguna Pindo (Ecuador, 1200 m) for the past 3000 yrs.

Establishing Transfer-Functions in this part of the world is very important and really novel (and much needed). Very little is known about (Holocene) TT changes because most of the available records are sensitive to precipitation. Indeed, the design of the TS has some problems (e.g. distribution of samples along the TT gradient) but, in

the real world, there is often not much one can do about this. A further shortcoming is that nutrients were not measured for the training set (e.g. Lotter et al 1998) and I would have done some technical details in a different way (sample uppermost 3-4 cm sediments instead on only 0-1 cm; exclude Long/Lat in the TF or test what happens if excluded). However, this is a first start and deserves publication. It might be worth testing whether the TF could be enhanced and optimized by removing stepwise lakes with extreme properties (such as e.g. the very shallow lake with 10 cm water depth).

The temperature reconstruction from the sediments of Laguna Pindo (c. 3000 yrs) stands on much weaker foundations and requires careful (major) revision, further testing and likely adding more samples to test the robustness and reproducibility of the cold spells. The conclusions of the current version of this paper (mainly the cold spells and their temperature) are barely supported by the data and remain speculative. The main challenges with the reconstruction:

Amplitudes for (multi)decadal (10-20 yrs according to the sample resolution) mean annual temperatures on the order of 4°C within the last 500 yrs and ca 7°C for late Holocene TT changes seem unrealistically high compared with what is known from other parts of the world including the tropics (Marcott et al. 2013; PAGES 2k 2013). The finding reported by Polissar et al. 2013 (inferred from dELAs of two glaciers in Venezuela at 4600 and 5000 masl) is an exception, and has been explained with very special local conditions at high elevation sites. Yet a plausible physical explanation is missing for the very large amplitudes of the cold spells found in Laguna Pindo (this manuscript). TT variability in the tropics are a very important issue and, thus, require much better foundations and support by data (including replications). I rather suspect (which should be explored/discussed by the authors) that the large amplitudes are related to problems with the TF, with the fact that many taxa in the fossil downcore samples of L Pindo are not represented in the calibration data set (! Line 495; I think this is a real problem), with the fact that Laguna Pindo is not well represented in the TS lakes (Line497), other variables (such as precipitation) play a major role particularly

in samples with poor fit-to-temperature (which is precisely the case in those samples with large TT amplitudes) and/or with some of the downcore samples (low numbers of hc, poor goodness-of-fit or poor modern analogues). These problems are honestly discussed in the text. It appears that L Pindo was not the best lake to perform a reconstruction (downcore analysis).

In my view (also your statement in Line 531) the TT reconstruction is, thus, rather qualitative than quantitative, it is not known what the (mixed) TT signal actually is. This might, in turn, explain the unrealistically high TT amplitudes of the L Pindo reconstruction.

For a publication in CP I would expect that a few additional samples should be analyzed to assess whether the prominent results (cold spells) are robust and can be reproduced or whether these single samples (e.g. at 20 cm sediment depth) could be artifacts, outliers or coincidence. This information is most relevant for the quality of the paper and the implications. Two examples:

(i) L335: The 'sudden drop at 1600 cal BP ' is inferred from just one (1) sample which has a substantial error (Fig. 7), poor goodness-of-fit (Fig 9), no good modern analogue (Fig. 9) and very low hc concentrations (Fig 9); the number of hc is not known. Is this really robust and significant?

(ii) L340: the short minimum around AD 1850: only one single sample at 20 cm; the error is substantial (Fig. 7), goodness-of-fit poor (Fig 9), no good modern analogue (Fig. 9) and very low hc concentrations (Fig 9); again, the number of hc is not known. This requires additional support by data from adjacent sediment samples or parallel cores. In general, counts (number of hc) should be given in all figures (in addition to hc concentrations). This is also important for the assessment of other parameters (e.g. L361-363) and the overall quality/robustness of the results.

Moreover, it is very important (and in my view conditional) that the raw data of the Training Set and TF (environmental variables and chironomids) as well as the raw data

of the chironmid stratigraphy od L Pindo are made available in digital form (as SOM to the publication).

In summary: The Transfer Function is important, although not perfect, but deserves publication. The L Pindo reconstructions has major deficits. I see two options: (i) The TF is optimized, undergoes additional testing, the quality of the reconstruction is substantially improved (robustness of the TT amplitudes, robustness of the cold anomalies, etc.) and/or (ii) the profile of the reconstruction is lowered; given the pertinent deficits the reconstruction is qualitative and NOT quantitative, not overstating the results and conclusions. I keep regarding this as publishable. All data (Transfer Function and reconstruction) should be made digitally available in a data repository. While this is standard in many scientific communities this does not yet seem to be the case with chironomids.

Specific Comments:

The Introduction could be shortened (quite lengthy).

Chapter 3: I would not make too many sub-chapters (only one paragraph in 3.4 and 3.5)

The sampling design for the downcore analysis should be described in detail (continuous, discrete sampling, regular intervals, stratigraphically...?) What is the percentage of sediment that is actually covered in the analysis? (e.g. 1 cm slice every 10 cm sediment makes 10% coverage and 90 % is not covered; this has serious implications regarding the robustness of the reconstruction).

I would include the Suppl Fig (Chronology) in the manuscript.

Title: reconcile. It is quantitative indeed, but how robust and how good are the numbers? => Qualitative

L77: Shulmeister

L93: ... preceding Glacial and Late-Glacial period ... (if you refer to 25-11.7 kBP; 20-25 kPB is not Late Glacial)

L95 ff: Make also reference to Marcott et al 2013. This is the most comprehensive dataset.

L99-100: Growing evidence from the tropics? I'm not sure about this. In fact it is still very controversial whether cold events (depending on the time scale) were globally, hemispherically or regionally synchronous (Wanner et al. 2011 QSR, Neukom et al 2014. NatCC;PAGES 2k 2013). The PAGES 2k Consortium 2013 has shown that, with a few exceptions (with solar-volcanic downturns) multi-decadal long cold phases were not coherent across the globe. Maybe rephrase sentence.

L108-109. References not appropriate (these are not climatology papers). Make reference to Garreaud et al 2009 or Stefan Hastenrath 1991 Climate Dynamics of the Tropics or similar.

L150. ... gradient of 24°C (not 25°C; from 0.8 to 25°C)

L151: How reliable are WTs in a 10 cm deep water body? It should be assessed how sensitive the TF is with/without such lakes. In such water bodies the difference between MAT and WT is typically very large (in particular Tmax). I guess that the TF stats could be improved.

L154: ... uppermost 1-2 cm ... representing 5-20 years... Well, it was done like this and is usually done like this. But this implies that the sample for the Training Set depicts in one lake interannual/subdecadal variability (which may be very different from climatology!) and in another sample it is rather climatology (20-30 yrs). I suspect that this adds substantial errors to the TF. Suggestion: if such large TT gradients are used (24°C) use the uppermost 3-4 cm of sediment to make sure that 20-30 yrs (climatology) are represented. The TT trends during 30 yrs are relatively small and similar in all lakes of the training set.

L171: Fourteen 14C samples? Fig SOM shows six of them. Where are the others? Pls change and make it consistent with L 324 ff.

L188: Were nutrients (N and P) not measured? This might be a problem (Lotter et al 1998 J Paleolimnology)

L272 and 275: avoid references in the results section. This reads like 'Discussion'

L280 . . . optimum. . . (?)

L295: in general, hc counts should be given in all Figures and Tables. L295 ff is rather Discussion than Results. Move this paragraph.

L300: Yes, this is critical (number of hc). It should be assessed whether the number of hc has an effect on the calibration statistics, in particular the residuals. (see also L303, I am not sure if this is the only criterion according to which the TF could be optimized)

L302: Table 1 does not show these details (which are important), Table 1 shows the summary only. The details (hc) should be given (in the SOM)

L327: The sampling design must be clarified (in the Methods section). You took 30 samples spread over 420 cm. How did you take the samples? 1 cm slice every 10-15 cm? Stratigraphically (according to which criteria?) or continuously (complete sediment section)?

L353: .. only seven samples? According to Fig 9 and the vertical dashed line there are many more.

L485: I don't think that anything is known about the precip/temperature relationship during the Late Holocene.

L 495: I think this is a substantial problem.

L531: according to this statement I would conclude that the temperature reconstruction of Laguna Pindo is qualitative at best.

L539: maybe also refer to Kanner et al (speleothems) and Ledru et al (N Ecuador)

L548: Jones & Mann 2004 is not the best (has been criticized; S-Hemisphere is very poor). Suggestion: PAGES 2k 2013.

L552/553: I don't think this is true. There's a large body of literature pointing out the role of volcanoes, or a combination of S+V . . . rephrase sentence.

L555: No, I don't think this is true (cool from 400 yr BP onwards). The sample at 250 yr BP is still among the warmest of the entire record, almost as warm as today (!). There is only 1 sample (at 1850 AD) that shows cool conditions, and it is very questionable how robust that is (see your comment and my comments above)

L569ff: It has been repeatedly demonstrated that the Andean ice cores (stable isotopes) record precipitation and not temperature (as claimed by Thompson et al).

L572: TT drop of 3-4°C during the LIA. Yes, this value has been reported for two Venezuelan glaciers (at 4600 and 5000 masl, mainly inferred from a drop in ELA by 300-500 m; Polissar et al. 2006). I doubt that similar (special high-elevation) conditions apply for L Pindo, given the limitations of the reconstruction (see above). This value seems extraordinarily high to me. Alternatively an explanation should be provided showing that such large TT amplitudes are physically plausible at local scales.

L598. Yes, the potential is shown (with the TF). But the reconstruction has major problems and severe limitations (see above). I would say: qualitative at best.

L605: . . .). Special. . .

L624: reference listed twice

L634: Dryas-Holocene

L667: check carefully

L668: . . .Science 289,

L680. Vol missing

L702: Lemke

L810: ... Science 234, ...

L814: Ref listed twice (also L819)

L818: Holocene

L841 Woodward, C.

L855: LOI: specify 550 or 950; ditto L858, Table 1 and Table 2, L883

Table 1: Data set should be made available in full detail

Table 3: add units (where appropriate), also Caption Fig 5

Fig 2: pH

Fig 3 (all Figs where appropriate, Fig 6, Fig 9): numbers of hc should be shown. It would be interesting to see the 'unusual lakes' (e.g. those with water depth of 10 cm).

---

## Author Comment (AC2) · 21 Mar 2016

Quantifying late-Holocene climate in the Ecuadorian Andes using a chironomid-based temperature inference model

We wish to thank everyone who contributed to the improvement of this manuscript, specifically the comprehensive suggestions from the two reviewers. Outlined below is a detailed response to both the reviewers' general, and specific comments. The manuscript has been substantially changed as a result of the reviewer's general comments (see below; response to general comments) and minor edits have been corrected and recorded accordingly. All typos and minor formatting errors, highlighted in the annotated pdf have also been changed.

Response to General comment: reliability of the reconstruction.

Whilst both reviewers, and an independent author who contributed to the online discussion, commented on the value of the study, all had major concerns relating to the final environmental reconstruction from Laguna Pindo and the subsequent interpretation. We would agree with all the reviewers that the development of a chironomid transfer function for the tropical Andes is an important contribution to tropical palaeolimnology and paleoclimatology. We also acknowledge, however, that chironomid studies from the tropics remain rare and little is known about the autecology of many of the taxa, namely their ecological tolerances relating to climatic variables. As a result, the environmental reconstruction from Laguna Pindo has some issues, namely unrealistically cold temperatures and significant inter sample variability. We would agree with all reviewers, and acknowledge in the original manuscript, that many of these fluctuations most likely relate to issues with the transfer function and/or the fossil record. These could include:

- The response of secondary variables, namely precipitation. - Low head capsule concentrations in many of the samples. - Un-even distribution of calibration lakes over the environmental gradient due to the steep topography of the Andes. - Taxonomic issues (i.e different species between fossil samples and modern samples that currently cannot be separated using only larval head capsule material).

Many of these limitations are discussed in the manuscript. Indeed, we would argue a central point of this work would be the comparison of WA and Bayesian methods, in order to further explore these limitations. The application of the Bayesian model results in a less variable reconstruction, and an explanation for why the uncertainty associated with the reconstruction is greater than the climate variability we are reconstructing. The individual likelihood function of fossil taxa and the resulting posterior probability distribution for temperature sheds light on how the un-even distribution of calibration lakes, and subsequent skewed distribution of taxa, is affecting the inferred temperatures. Many taxa have unrealistically cold temperature optima due to the over

representation of cold lakes in the calibration datasets and this has a significant affect on the reconstruction, most notably unrealistic cold Holocene temperatures. The error associated with both reconstructions is entirely consistent with a constant temperature of 20°C. We would agree with the reviewers that attributing the variability of the reconstruction to anything more than noise would be an overstatement at this point. This work does, however, suggest the way forward for improving temperature reconstructions, namely, improving the richness/sampling of the training set to enable the detection of smaller signals.

For this reason we would agree with the anonymous reviewer; " These problems are honestly discussed in the text. It appears that L.Pindo was not the best lake to perform a reconstruction.". The reviewer provides two options for rectifying these issues:

"(i) The TF is optimized, undergoes additional testing, the quality of the reconstruction is substantially improved (robustness of the TT amplitudes, robustness of the cold anomalies, etc.) and/or (ii) the profile of the reconstruction is lowered; given the pertinent deficits the reconstruction is qualitative and NOT quantitative, not overstating the results and conclusions."

Unfortunately, we do not feel that the quality of the reconstruction can be substantially improved at this stage for many of the reason discussed previously and therefore we cannot meet first criteria (i). For this reason we propose to move forward with the reviewers second suggestion, i.e. lowering the profile of the reconstruction in the manuscript. We feel this option will allow the manuscript to make a meaningful contribution to the literature, whilst honestly representing the current sate of chironomid research in the area and addressing many of the concerns of all reviewers and online contributions.

The following major changes have been made to the manuscript in order to address the general comments of both reviewers:

i) The Introduction has been shortened (L.126-L132) in order to reflect the new focus of the manuscript, i.e. refining the proxy as opposed to palaeoclimate inferences. This modification also addresses a concern of the anonymous review, which noted the introduction as being overly long.

ii) Sub section 5.4 Laguna Pindo temperature reconstructions has been removed. This subsection is no longer needed as the temperature reconstruction is presented as qualitative and only used to further understand the various models.

iii) Sections 5.6 (Cooling climate 3800-2800 cal yrs BP), and 5.7 (Recent cooling) have been removed. Based on the recommendations of the reviewers, we have changed the focus of the manuscript to center on proxy development not palaeoclimate interpretations. The Laguna Pindo reconstruction is used to understand the limitations of each model and is presented as a more qualitative interpretation of climate variability over this time period. The conclusions of the manuscript focus on our future recommendations for improving palaeotemperature inferences using chironomids. This addresses the current limitations of Neotropical palaeolimnology using chironomids, and provides a list of necessary criteria for future researchers wishing to explore this proxy further.

The attached PDF provides a detailed breakdown of how each of the reviewers specific comments was addressed.

Please also note the supplement to this comment:
http://www.clim-past-discuss.net/cp-2015-186/cp-2015-186-AC2-supplement.pdf

**Supplement:**

**Response to reviewers**

***Quantifying late-Holocene climate in the Ecuadorian Andes using a chironomid-based temperature inference model***

We wish to thank everyone who contributed to the improvement of this manuscript, specifically the comprehensive suggestions from the two reviewers. Outlined below is a detailed response to both the reviewers' general, and specific comments. The manuscript has been substantially changed as a result of the reviewer's general comments (see below; *response to general comments*) and minor edits have been corrected and recorded accordingly. All typos and minor formatting errors, highlighted in the annotated pdf have also been changed.

**Response to General comment: *reliability of the reconstruction*.**

Whilst both reviewers, and an independent author who contributed to the online discussion, commented on the value of the study, all had major concerns relating to the final environmental reconstruction from Laguna Pindo and the subsequent interpretation. We would agree with all the reviewers that the development of a chironomid transfer function for the tropical Andes is an important contribution to tropical palaeolimnology and paleoclimatology. We also acknowledge, however, that chironomid studies from the tropics remain rare and little is known about the autecology of many of the taxa, namely their ecological tolerances relating to climatic variables. As a result, the environmental reconstruction from Laguna Pindo has some issues, namely unrealistically cold temperatures and significant inter sample variability. We would agree with all reviewers, and acknowledge in the original manuscript, that many of these fluctuations most likely relate to issues with the transfer function and/or the fossil record. These could include:

- The response of secondary variables, namely precipitation.
- Low head capsule concentrations in many of the samples.
- Un-even distribution of calibration lakes over the environmental gradient due to the steep topography of the Andes.
- Taxonomic issues (i.e different species between fossil samples and modern samples that currently cannot be separated using only larval head capsule material).

Many of these limitations are discussed in the manuscript. Indeed, we would argue a central point of this work would be the comparison of WA and Bayesian methods, in order to further explore these limitations. The application of the Bayesian model results in a less variable reconstruction, and an explanation for why the uncertainty associated with the reconstruction is greater than the climate variability we are reconstructing. The individual likelihood function of fossil taxa and the resulting posterior probability distribution for temperature sheds light on how the un-even distribution of calibration lakes, and subsequent skewed distribution of taxa, is affecting the inferred temperatures. Many taxa have unrealistically cold temperature optima due to the over representation of cold lakes in the calibration datasets and this has a significant affect on the reconstruction, most notably unrealistic cold Holocene temperatures. The error associated with both reconstructions is entirely consistent with a constant temperature of 20°C. We would agree with the reviewers that attributing the variability of the reconstruction to anything more than noise would be

an overstatement at this point. This work does, however, suggest the way forward for improving temperature reconstructions, namely, improving the richness/sampling of the training set to enable the detection of smaller signals.

For this reason we would agree with the anonymous reviewer; " *These problems are honestly discussed in the text. It appears that L.Pindo was not the best lake to perform a reconstruction.".* The reviewer provides two options for rectifying these issues:

"*(i) The TF is optimized, undergoes additional testing, the quality of the reconstruction is substantially improved (robustness of the TT amplitudes, robustness of the cold anomalies, etc.) and/or (ii) the profile of the reconstruction is lowered; given the pertinent deficits the reconstruction is qualitative and NOT quantitative, not overstating the results and conclusions.*"

Unfortunately, we do not feel that the quality of the reconstruction can be substantially improved at this stage for many of the reason discussed previously and therefore we cannot meet first criteria (i).  For this reason we propose to move forward with the reviewers second suggestion, i.e. lowering the profile of the reconstruction in the manuscript. We feel this option will allow the manuscript to make a meaningful contribution to the literature, whilst honestly representing the current sate of chironomid research in the area and addressing many of the concerns of all reviewers and online contributions.

The following major changes have been made to the manuscript in order to address the general comments of both reviewers:

i)      The Introduction has been shortened (L.126-L132) in order to reflect the new focus of the manuscript, i.e. refining the proxy as opposed to palaeoclimate inferences. This modification also addresses a concern of the anonymous review, which noted the introduction as being overly long.

ii)     Sub section 5.4 *Laguna Pindo temperature reconstructions* has been removed. This subsection is no longer needed as the temperature reconstruction is presented as qualitative and only used to further understand the various models.

iii)    Sections 5.6 (*Cooling climate 3800-2800 cal yrs BP)*, and 5.7 (*Recent cooling)* have been removed. Based on the recommendations of the reviewers, we have changed the focus of the manuscript to center on proxy development not palaeoclimate interpretations. The Laguna Pindo reconstruction is used to understand the limitations of each model and is presented as a more qualitative interpretation of climate variability over this time period. The conclusions of the manuscript focus on our future recommendations for improving palaeotemperature inferences using chironomids. This addresses the current limitations of Neotropical palaeolimnology using chironomids, and provides a list of necessary criteria for future researchers wishing to explore this proxy further.

**Response to specific comments**

| Line Number | Reviewers comment | Response |
|---|---|---|
| L. 38 R2 | **General Comment** Due to the limitations of the environmental reconstruction the anonymous review suggest reference to the reconstruction should not be "quantitative" | "…*the first quantitative reconstruction…*" has been removed |
| L. 349 R1 | I do not think that that lakes located between 1000 and 300 m asl can be considered to be similar in elevation to Laguna Pindo (~ 1200 m asl). Applying a standard lapse rate to this elevation range suggests that MAT for the lowest and highest lakes would vary by 12-20°C. | "…*although the samples plot within the range of modern calibration lakes that lie at similar elevations (1000-3000 m a.s.l).*" has been removed |
| L. 415 R1 | Reporting the RMSEP as a % of the total MAT range captured by the training set would be useful. | Sentence becomes; "*Although both models (WA inverse and Bayesian) perform well (WA RMSEP= 2.4°C/ 9.6% of training set range and Bayesian RMSEP= 2.3°C/9.2% of training set range)…*" |
| Fig 1 R1 | requires a N-arrow | N arrow has been added |
| Fig 2 R1 | "PH" should be corrected. | PH has been corrected to pH |
| R1 | It is not clear why non-limnological variables such as latitude were included in the exploratory analysis. Latitude, longitude and elevation are not directly controlling the distribution of midges; the analyses should be re-run with only environmental variables that have the potential to directly control the distribution of midges included. | These variables have been excluded from the analysis. |
| R2 | The Introduction could be shortened (quite lengthy). | L.126-L132 have been removed from the introduction and the manuscript shortened to reflect the new direction of the paper. |
| R2 | Chapter 3: I would not make too many sub-chapters (only one paragraph in 3.4 and 3.5) | Sub headings 3.4; 3.5; 3.6; and 3.7 have been removed |

| R2# | The sampling design for the downcore analysis should be described in detail (continuous, discrete sampling, regular intervals, stratigraphically. . .?) What is the percentage of sediment that is actually covered in the analysis? (e.g. 1 cm slice every 10 cm sediment makes 10% coverage and 90 % is not covered; this has serious implications regarding the robustness of the reconstruction). | Sentence added; *"The sampling interval for chironomid analysis was not uniform due to a varied sedimentation rate a varying sedimentation rate. To achieve as even a coverage possible over the time interval, samples were taken between every 10 and 20cm."* |
|---|---|---|
| R2 | I would include the Suppl Fig (Chronology) in the manuscript. | Table S1 and Figure S1 have know been included in the manuscript. |
| R2 | Title: reconcile. It is quantitative indeed, but how robust and how good are the numbers? => Qualitative | Title has been changed to; Inferring late-Holocene climate in the Ecuadorian Andes using a chironomid-based temperature inference model |
| L.77 R2 | Shulmeister | Corrected |
| L.93 R2 | ... preceding Glacial and Late-Glacial period ... (if you refer to 25-11.7 kBP; 20-25 kPB is not Late Glacial) | Changed to; *"(... c. 15,000-11,700 years before present...)"* |
| L.95 R2 | Make also reference to Marcott et al 2013. This is the most comprehensive dataset. | Marcott et al 2013 has been added. |
| L.99-100 R2 | Growing evidence from the tropics? I'm not sure about this. In fact it is still very controversial whether cold events (depending on the time scale) were globally, hemispherically or regionally synchronous (Wanner et al. 2011 QSR, Neukom et al 2014. NatCC;PAGES 2k 2013). The PAGES 2k Consortium 2013 has shown that, with a few exceptions (with solar-volcanic downturns) multi-decadal long cold phases were not coherent across the globe. Maybe rephrase sentence. | Changed to; "Some evidence from the tropics suggests Holocene climate fluctuations such as the LIA are maybe global events...." |
| L.108-109 R2 | References not appropriate (these are not climatology papers). Make ref- erence to Garreaud et al 2009 or Stefan Hastenrath 1991 Climate Dynamics of the Tropics or similar. | References have been modified. |
| L.151 | How reliable are WTs in a 10 cm deep water | In producing the manuscript we ran |

| R2 | body? It should be assessed how sensitive the TF is with/without such lakes. In such water bodies the difference between MAT and WT is typically very large (in particular Tmax). I guess that the TF stats could be improved. | the transfer function using multiple combinations of different lakes included and excluded. This included removing very shallow lakes and overly deep lakes. The results presented are for the best performing inference model. We believe the problems which are leading to the unreliable reconstruction are overwhelmingly those discussed with reference to the all reviewers general comments. The manuscript has been changed accordingly to address this. |
|---|---|---|
| L.154 R2 | . . . uppermost 1-2 cm . . . representing 5-20 years. . . Well, it was done like this and is usually done like this. But this implies that the sample for the Training Set depicts in one lake interannual/subdecadal variability (which may be very different from climatology!) and in another sample it is rather climatology (20-30 yrs). I suspect that this adds substantial errors to use the uppermost 3-4 cm of sediment to make sure that 20-30 yrs (climatology) are represented. The TT trends during 30 yrs are relatively small and similar in all lakes of the training set. | As pointed out by the reviewer the sampling method adopted here is common practice for chironomid studies of this kind. We would agree with the reviewer that testing the results of various sampling methods would be a worthy endeavor. The reviewer makes an important point that the uppermost sediments likely reflect inter-annual variability whilst deeper homogenous sampling may more accurately reflect climatology. Addressing this directly, however, would call for a complete re-sampling of the entire calibration dataset and will very probably not address the central concern of the reviewers; improving the reconstruction. This suggestion would not reduce the problems associated with un-even sampling. |
| L.171 R2 | Fourteen 14C samples? Fig SOM shows six of them. Where are the others? Pls change and make it consistent with L 324 ff. | The Laguna Pindo record is much older and longer than the portion presented here. Much of the record is radiocarbon infinite and work is on-going to produce a complete age depth model. Presented here is the portion of the record for which chironomid remains are found. This is addressed in Line 330-332 *"The best-fit age depth model for Laguna Pindo was a smooth spline (Fig S1). Due to the absence of chironomids at the bottom of the sequence, six radiocarbon samples were used for building the model with a total depth of the sediment considered of 461 cm"*. |

| | | This figure has been removed from the SOM and placed in the manuscript itself. |
|---|---|---|
| L188 R2 | Were nutrients (N and P) not measured? This might be a problem (Lotter et al 1998 J Paleolimnology) | Samples were taken for nutrients (anions and cation). Although filtered in the field, due to the remoteness of the fieldwork and continued biological activity, these samples were no longer reliable once returned for laboratory analysis in the UK. |
| L272 and 275 R2 | avoid references in the results section. This reads like 'Discussion' L280 . . . optimum. . . (?) | References have been removed

*"…optima"* changed to *"…optimum"* |
| L295 R2 | in general, hc counts should be given in all Figures and Tables. | Hc counts has been added to all figures. The total number of head capsules for each calibration lake can be found in the data archive or *Matthews-bird et al 2015* |
| L295 f | is rather Discussion than Results. Move this paragraph. | Paragraph has been moved to discussion |
| L300 R2 | Yes, this is critical (number of hc). It should be assessed whether the number of hc has an effect on the calibration statistics, in particular the residuals. (see also L303, I am not sure if this is the only criterion according to which the TF could be optimized) | We agree that the affect of head capsule concentration is extremely important, particularly with regard to WA models, which rely heavily on abundance. The Bayesian model, however, has a component of the model that uses only presence absence data. This was one reason for comparing the two methods. The Bayesian reconstruction and likelihood function, shows the effect of head capsule concentration on the reconstruction. Particularly the bias towards to colder temperatures. The current methodology already addresses the concern of the reviewer. |
| L.302 R2 | Table 1 does not show these details (which are important), Table 1 shows the summary only. The details (hc) should be given (in the SOM) | Total number of head capsules for each lake can be found in the data repository. |
| L.327 R2 | The sampling design must be clarified (in the Methods section). You took 30 samples spread over 420 cm. How did you take the samples? 1 cm slice every 10-15 cm? Stratigraphically (according to which criteria?) or continuously (complete sedi-ment section)? | This has been addressed by a previous comment R2#. |
| L.353 R2 | .. only seven samples? According to Fig 9 and the vertical dashed line there are many | This was a typo that has been rectified, 14 samples have a poor fit |

| | more. | to temperature. |
|---|---|---|
| L.485
R2 | I don't think that anything is known about the precip/temperature relationship during the Late Holocene. | Sentence changed to;

*"The location of Laguna Pindo makes it a good palaeoecological setting to record the response of temperature-sensitive proxies"* |
| L.495
R2 | I think this is a substantial problem. | We agree with the reviewer that the lack of modern analogues is a substantial problem with the reconstruction. This is honestly discussed in the paragraph cited. This lack of modern analogues most likely reflects the uneven distribution of calibration lakes and the particular lack of lakes surrounding the fossil site. |
| L. 531
R2 | according to this statement I would conclude that the temperature reconstruction of Laguna Pindo is qualitative at best. | We agree with the reviewer that more work is needed before Neotropical chironomids can be described as quantitative. As highlighted in our response to the general comments we accept that the profile of the reconstruction should be lowered. The passage now reads;

*"The WA inverse MAT reconstruction, however, is statistically significant based on the criteria described by Telford and Birks (2011a) (Fig 10) suggesting that despite conflicting variables a temperature signal can be obtained from Neotropical chironomids although we caution against an over interpretation at this stage. Due to some of the limitations discussed previously, the reconstruction can currently only be deemed qualitative and requires more research."* |
| L. 539
R2 | maybe also refer to Kanner et al (speleothems) and Ledru et al (N Ecuador) | This section has now been removed and significantly modified. The manuscript no longer over interprets the final reconstruction and these suggestions are no longer relevant. |

| | | |
|---|---|---|
| L548 R2 | Jones & Mann 2004 is not the best (has been criticized; S-Hemisphere is very poor). Suggestion: PAGES 2k 2013. | See previous comment |
| L.552/553 R2 | I don't think this is true. There's a large body of literature pointing out the role of volcanoes, or a combination of S+V . . . rephrase sentence. | See previous comments |
| L.555 R2 | No, I don't think this is true (cool from 400 yr BP onwards). The sample at 250 yr BP is still among the warmest of the entire record, almost as warm as today (!). There is only 1 sample (at 1850 AD) that shows cool conditions, and it is very questionable how robust that is (see your comment and my comments above) | See previous comment |
| L.569ff R2 | It has been repeatedly demonstrated that the Andean ice cores (stable iso- topes) record precipitation and not temperature (as claimed by Thompson et al). | See previous comment |
| | LIA. Yes, this value has been reported for two Venezuelan glaciers (at 4600 and 5000 masl, mainly inferred from a drop in ELA by 300-500 m; Polissar et al. 2006). I doubt that similar (special high-elevation) conditions apply for L Pindo, given the limitations of the reconstruction (see above). This value seems extraordinarily high to me. Alternatively an explanation should be provided showing that such large TT amplitudes are physically plausible at local scales. | See previous comment |
| L.598 R2 | Yes, the potential is shown (with the TF). But the reconstruction has major problems and severe limitations (see above). I would say: qualitative at best. | We agree with the reviewer and have modified the manuscript accordingly |
| L.605 R2 | : . . .). Special. . . | Rectified |
| L624 R2 | reference listed twice L634: Dryas-Holocene L667: check carefully L668: . . .Science 289 | Duplicate reference removed |
| L.680. R2 | Vol missing | Volume added |
| L.702 R2 | Lemke | Rectified |
| L.810 R2 | ... Science 234, ... | Rectified |

| L.814 R2 | Ref listed twice (also L819) | Duplicate reference removed |
|---|---|---|
| L. 818 R2 | Holocene | Rectified |
| L.841 R2 | Woodward, C | Rectified |
| L.855 R2 | LOI: specify 550 or 950; ditto L858, Table 1 and Table 2, L883 | 550 has been sepcified |
| Table 1 | Data set should be made available in full detail | Data is now available at Data Dryad |
| Table 3: | add units (where appropriate), also Caption Fig 5 | Units added |
| Fig 2: | pH | PH changed to pH |
| Fig 3 | (all Figs where appropriate, Fig 6, Fig 9): numbers of hc should be shown. It would be interesting to see the 'unusual lakes' (e.g. those with water depth of 10 cm). | Hc has been added to all necessary figures; fig 3,6,9 |

---

## Author Response (AR2)

Response to referees: cp-2015-186

Quantifying late-Holocene climate in the Ecuadorian Andes using a chironomid-based temperature
inference model

Frazer Matthews-Bird; Stephen J. Brooks; Philip B. Holden; Encarni Montoya, and William D. Gosling

We wish to thank all the reviewers involved in the preparation of this manuscript, and the editorial board of *Climate of the Past* for the opportunity to submit a revised version of the original manuscript. We feel the extended review process has resulted in a superior study, which will appeal to a broad readership and make a significant contribution to the field. Below is a detailed breakdown of how the latest round of reviewer comments have been addressed.

Referee #1: *I would like to commend the authors for thoroughly engaging with the reviewers' comments and making substantive changes to the ms. I believe the ms greatly improved and deserving of publication. I look forward to seeing the ms published.*

Anonymous Referee #2: comment 1
*The authors have addressed the issues adequately. The changes are well documented. However, in accordance with the revised text body, also the Abstract must be modified (mention the caveats of the TF and the reconstruction in L Pindo) in the sense that the reconstruction is qualitative; (e.g. add 1-2 sentences Line 487 – 489 and/or Lines 687-689).*

Response to Referee #2: comment 1
We have modified the abstract in line with the reviewer's comments. Lines 20-23 now reads:
*We would caution, however, against an over interpretation at this stage. The reconstruction can only currently be deemed qualitative and requires more research before quantitative estimates can be generated with confidence.*

Anonymous Referee #2: comment 2
*Maybe I have missed it in the text, but it should be mentioned somewhere that all data (Transfer Function and Downcore stratigraphy) are publicly available at www.xy (NERC website). This is very important.*

Response to Referee #2: comment 2
The data is stored with the National Geoscience Data Centre (NGDC) and can be found at http://www.bgs.ac.uk/downloads/home.html. The data repository process can take 3 months to finalize before a DOI is issued, however we understand the data can be accessed using the repository search engine. This link has now been included in the acknowledgments, however, we would also be happy to make the data available as supplementary information. We leave that decision to the editorial board.

Anonymous Referee #2: comment 3
*It's a question of style, but Sections 4.1 and 4.3 are very short (just two sentences or so). Maybe combine L411: 929 cm.*

Response to Referee #2: comment 2
Section 4: Results now has just three subsections in line with reviewer's comments.
4. Results
    4.1 Calibration data set
    4.2 Laguna Pindo fossil chironomids and dating.
    4.3 Palaeotemperature reconstructions

[revised manuscript text omitted]

**Figures**

**Figure 1**

[Figure]

Figure 2

[Figure]

**Figure 3**

[Figure]

frazer 3/24/2016 10:06 AM

**Comment [4]:** Latitude removed from the analysis frazer 2/26/2016 3:44 PM

frazer 2/26/2016 2:15 PM

Unknown

Unknown

**Figure 4**

frazer 2/26/2016 3:44 PM

frazer 2/26/2016 2:15 PM

Unknown

[Figure]

**Figure 5**

frazer 2/26/2016 3:44 PM

[Figure]

Mean Annual Temperature (MAT ℃)

**Figure 6**

frazer 2/26/2016 3:44 PM

[Figure]

**Figure 7**

frazer 2/26/2016 3:44 PM

[Figure]

Unknown

**Figure 8**

frazer 2/26/2016 3:44 PM

Unknown

[Figure]

**Figure 9**

[Figure]

frazer 2/26/2016 3:44 PM

frazer 2/25/2016 2:42 PM

**Figure 10**

[Figure]

**Figure 11**

[Figure]

frazer 2/26/2016 3:44 PM

**Figure 12**

frazer 2/26/2016 3:45 PM